# The binding of Borealin to microtubules underlies a tension independent kinetochore-microtubule error correction pathway

Prasad Trivedi[1], Anatoly V. Zaytsev[2,3], Maxim Godzi[3], Fazly I. Ataullakhanov[3,4,5], Ekaterina L. Grishchuk [2,4] & P. Todd Stukenberg[1,6]

Proper chromosome segregation depends upon kinetochore phosphorylation by the Chromosome Passenger Complex (CPC). Current models suggest the activity of the CPC decreases in response to the inter-kinetochore stretch that accompanies the formation of bi-oriented microtubule attachments, however little is known about tension-independent CPC phosphoregulation. Microtubule bundles initially lie in close proximity to inner centromeres and become depleted by metaphase. Here we find these microtubules control kinetochore phosphorylation by the CPC in a tension independent manner via a microtubule-binding site on the Borealin subunit. Disruption of Borealin-microtubule interactions generates reduced phosphorylation of prometaphase kinetochores, improper kinetochore-microtubule attachments and weakened spindle checkpoint signals. Experimental and modeling evidence suggests that kinetochore phosphorylation is greatly stimulated when the CPC binds microtubules that lie near the inner centromere, even if kinetochores have high inter-kinetochore stretch. We propose the CPC senses its local environment through microtubule structures to control phosphorylation of kinetochores.

[1] Department of Cell Biology, University of Virginia, Charlottesville 22908 VA, USA. [2] Department of Physiology, Perelman School of Medicine, University of Pennsylvania, Philadelphia 19104 PA, USA. [3] Center for Theoretical Problems of Physicochemical Pharmacology, Russian Academy of Sciences, Moscow 119991, Russia. [4] National Research Center of Pediatric Hematology, Oncology and Immunology, Moscow 117198, Russia. [5] Moscow Institute of Physics and Technology, Moscow 141701, Russia. [6] Department of Biochemistry and Molecular Genetics, University of Virginia, Charlottesville 22908 VA, USA. Correspondence and requests for materials should be addressed to E.L.G. (email: gekate@pennmedicine.upenn.edu) or to P.T.S. (email: pts7h@virginia.edu)

Human kinetochores bind ~20 microtubules and faithful chromosome segregation requires that the majority of the microtubules attached to one sister kinetochore orient towards one spindle pole, while those of its sister orient towards the opposite pole (biorientation)[1]. The inability to obtain biorientation is a major source of chromosomal instability in tumors[2,3]. The Chromosome Passenger Complex (CPC), a four-protein complex consisting of chromatin targeting subunits Survivin and Borealin, the scaffold INCENP and a kinase Aurora-B, controls biorientation as well as other mitotic events by phosphorylating kinetochore substrates and destabilizing kinetochore-microtubule attachments[4]. The majority of the CPC (~75%) is localized to the inner-centromere, which is the chromatin between kinetochores on mitotic chromosomes, during prometaphase and metaphase[5,6]. Inner centromere localization is believed to concentrate the protein to enable kinase auto-activation[7]. CPC recognizes the inner centromere via two distinct histone phosphorylation marks, Histone H3 phosphorylated on T3 (H3pT3)[8–10] and Histone H2A phosphorylated on T120 (H2ApT120)[4,8,11–14]. The CPC phosphorylates kinetochore substrates that are greater than 500 nm away from inner centromeres[15,16]. Phosphorylation of kinetochore substrates such as the Ndc80 complex, by Aurora-B, is higher on unaligned kinetochores than metaphase-aligned kinetochores[15,17], which may regulate many events including the maturation of kinetochore-microtubule attachments[18]. This is caused in part by recruitment of phosphatases to kinetochores after they obtain proper kinetochores attachments[19–21], but most models suggest that the CPC's ability to phosphorylate kinetochores is also decreased in metaphase[22–24].

How the CPC phosphorylates kinetochores and why kinetochore phosphorylation is higher in unaligned chromosomes than aligned chromosome is a matter of intense research. It has been proposed that centromere anchored CPC uses an extended single alpha-helix (SAH) on the INCENP subunit to reach the kinetochore substrates and phosphorylate them[22,23]. Upon biorientation the pulling force exerted by the kinetochore bound microtubules increases the distance between the CPC and its kinetochore-localized substrates thus reducing the INCENP's reach and therefore phosphorylation of kinetochore substrates. Another model suggests that the centromeric pool of the CPC activates soluble CPC that propagates to kinetochores via a reaction-diffusion mechanism that involves chromatin-bound CPC[24,25]. A pool of the CPC may directly localize to kinetochores[22,26], however, direct binding of kinetochores is unlikely to be the only mechanism because depletion of the centromere-bound pool or expression of CPC mutants that do not localize to inner centromeres compromises the ability of Aurora-B to phosphorylate distant substrates[24,25,27]. Budding yeast and chicken DT40 cells do not require centromere localization for biorientation[28–30], but the CPC in yeast require the ability to bind microtubules[28,29].

Many of these models suggest that the CPC is regulated by changes to the inner centromeric chromatin that results from the pulling forces exerted by microtubules bound to the kinetochores (inter-kinetochore stretch or centromeric tension)[22,31,32]. Apart from tension sensitive mechanisms, the tension-independent mechanisms are also likely to be involved since some pro-metaphase kinetochores may also become stretched due to kinetochore localized motor activity on microtubule bundles that lie in close proximity to inner centromeres[33,34].

It was recently shown that the initial kinetochore-microtubule attachments in prometaphase place the inner-centromere regions adjacent to large bundles of microtubules that also run adjacent to sister kinetochores[33]. These observations suggested that there is distinct prometaphase state when inner centromeres are in close proximity with spindle microtubules that span from inner-centromeres to kinetochores and beyond. These inner centromere proximal microtubules are largely reduced by metaphase[33] when they are replaced by the "end-on" attachments of mature kinetochore fibers (K-fibers). Moreover, the CPC was also shown to localize specifically to these centromere proximal microtubules in prometaphase[35]. Microtubules stimulate the CPC activity and auto-activation in vitro, and they are required for proper localization of the CPC to the inner-centromere[35–37]. Microtubules are also required for full activation of the CPC in a *Xenopus* extract system where the clustering of CPC by chromatin is replaced by activation by dimerizing antibodies[38]. The SAH domain of INCENP binds microtubules and is important for the maintenance of the paclitaxel-dependent SAC arrest[37,39,40]. However, it is unclear whether this region is required to correct improper kinetochore-microtubule attachments[37,41].

Here, we investigated the role of CPC-microtubule interaction in regulation of kinetochore phosphorylation. Specifically, we have identified a microtubule-binding site in the Borealin subunit of the CPC, and show that this interaction is important for kinetochore phosphorylation, proper error-correction and for maintenance of paclitaxel-induced SAC-dependent arrest. Using theoretical approaches, we demonstrate that microtubules near the inner centromere stimulate kinase activity at adjacent kinetochores. Interestingly, microtubules extend CPC activity even further than kinetochores at their most stretched state, suggesting that the regulation by microtubules is independent of chromatin changes by tension. Our model also demonstrates that end-on attached K-fiber microtubules do not stimulate phosphorylation of kinetochores to the same extent. Together these findings uncover a tension-independent layer of kinetochore regulation where local changes to microtubules near inner centromeres stimulate kinetochore phosphorylation by the CPC.

## Results

**Borealin contains a microtubule (MT)-binding site**. The centromere-targeting region of human CPC was expressed and purified from *E. coli*. Our preparations contained the first 48 amino acids of INCENP tagged on the N-terminus with 6-histidines, and full-length Survivin and Borealin (ISB)). ISB was mixed with paclitaxel-stabilized microtubules and the protein that remained bound to microtubules after sedimentation was quantified by immunoblot. The ISB sub-complex bound taxol-stabilized microtubules with an apparent Kd of ~164 nM (Fig. 1a, b). We also observed microtubule binding events with Total Internal Reflection Fluorescence (TIRF) Microscopy of single GFP-ISB molecules on taxol-stabilized microtubules (Fig. 1c, Supplementary Fig. 1C). GFP-ISB randomly diffused on taxol-stabilized microtubules with diffusion coefficient $0.73 \pm 0.09\ \mu m^2/s$ and residence time 0.41 s (Supplementary Fig. 1C–E). Proteins often bind microtubules through the electrostatic interaction between basic amino acids in proteins and the acidic residues on the E-hook of tubulin subunits. There are evolutionarily conserved basic residues at the tip of the triple helix of the ISB structure (Fig. 1d, e). Disruption of these basic residues by expression of Borealin$^{R17E, R19E, K20E}$ was previously shown to inhibit cytokinesis, but no effects on early mitotic events were reported[42]. We engineered these mutations into our *E. coli* expression construct to generate ISB$^{MTBD}$. These charge reversal mutations in the basic patch dramatically reduced the microtubule binding affinity of the ISB complex (apparent Kd of 4200 nM) (Fig. 1a, b). Deleting the N-terminal 20 amino acids of Borealin (ISB$^{\Delta 20}$) also reduced the affinity for microtubule binding (apparent Kd of 750 nM, Fig. 1a, b). These mutations did not hinder ISB complex formation or caused any gross structural

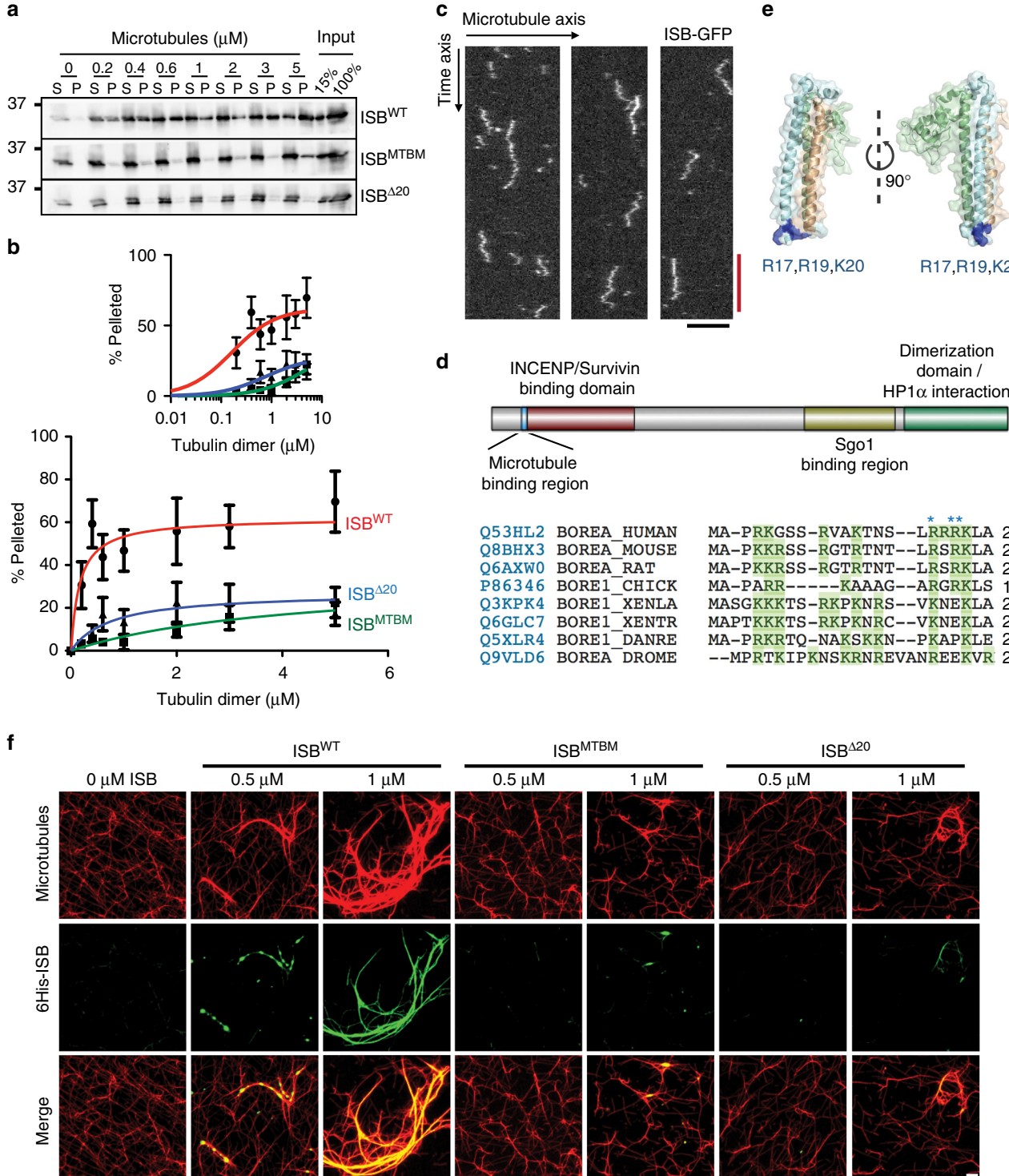

**Fig. 1** Borealin binds microtubules through its N-terminal region. **a** Western blots of input, supernatant (S) and pellet (P) fraction of microtubule co-sedimentation assay with 100 nM ISB^WT, ISB^MTBM and ISB^Δ20 and indicated concentration of microtubules; probed with anti-Borealin antibody. **b** Graph from three independent microtubule co-sedimentation assays (mean ± s.d.), small graph is $\log_{10}$ scale and large graph is linear scale. ISB^WT is in red, ISB^MTBM is in green and ISB^Δ20 is in blue. **c** Representative kymographs (from two independent experiments with 150 tracks) of single molecules of ISB-GFP diffusing on taxol-stabilized microtubules, ISF-GFP concentration 100 pM. Black scale bar 5 μm, red scale bar 1 s. **d** Schematic showing multiple protein interaction regions on Borealin (microtubule-binding region is characterized in this paper). Multiple sequence alignment of Borealin, basic residues are shown in green, residues important for microtubule binding are indicated with blue asterisk. **e** Crystal structure of INCENP-survivin-Borealin (PDB: 2QFA); microtubule binding residues R17, R19, and K20 are highlighted in dark blue. **f** Images from microtubule bundling assay ISB^WT, ISB^MTBM and ISB^Δ20 were incubated with 1 μM taxol-stabilized microtubules and probed with anti-tubulin and anti-6His antibody. 6His-ISB is shown in green and microtubules are shown in red (representative images from one of two independent experiments). Scale bar is 5 μm

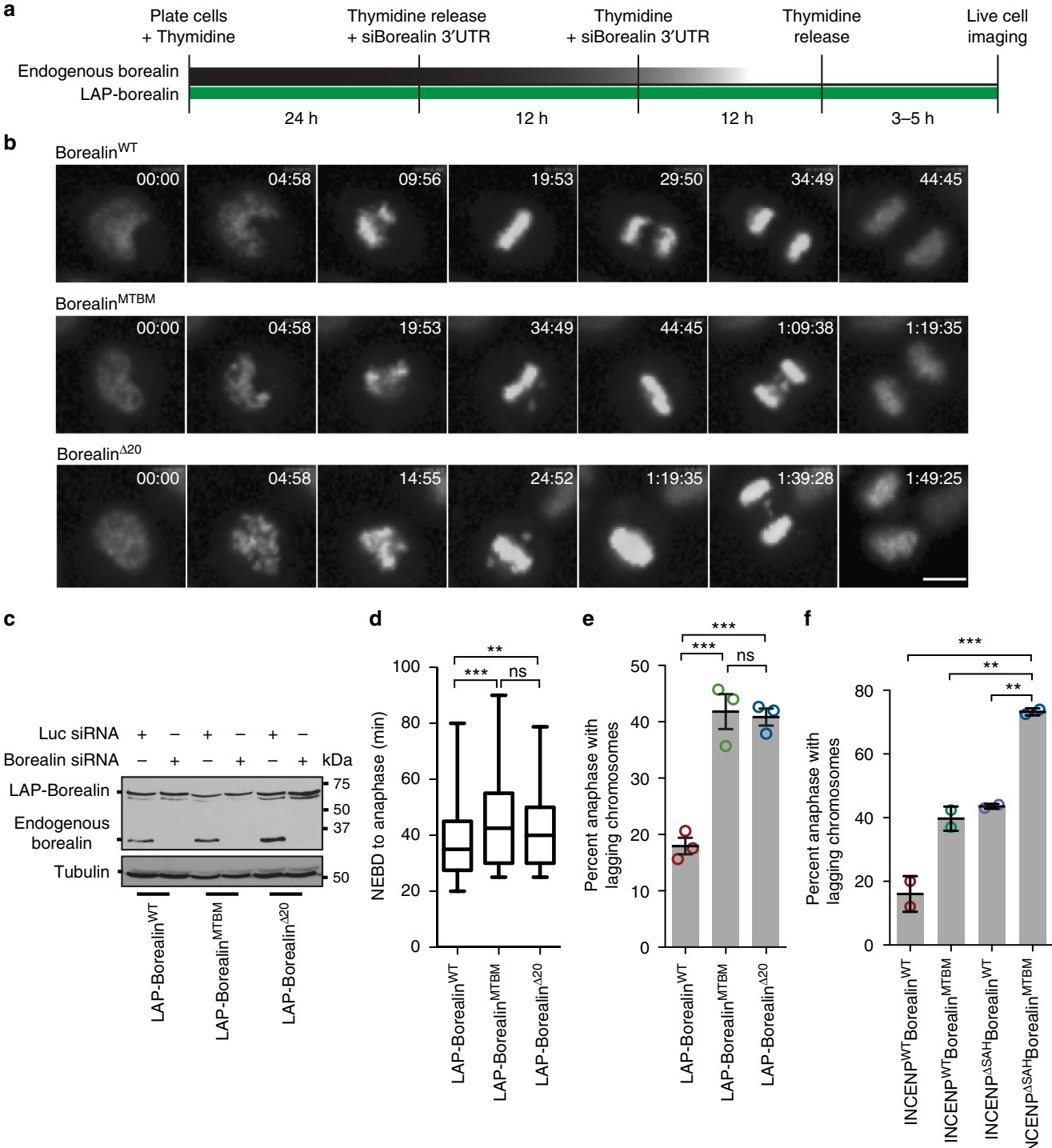

changes as measured by gel-filtration chromatography (Supplementary Fig. 1A, B). We immunoprecipitated tagged Borealin$^{MTBM}$ or Borealin$^{WT}$ that was expressed in HeLa cells and pulled down similar amounts of endogenous untagged Borealin demonstrating that the mutant does not affect dimerization (Supplementary Fig. 1G).

ISB$^{WT}$ complex also bundled paclitaxel-stabilized microtubules in a concentration-dependent manner (Fig. 1f). Interestingly ISB specifically bound the bundled microtubules; both ISB$^{MTBD}$ and ISB$^{Δ20}$ were deficient in the bundling of paclitaxel-stabilized microtubules (Fig. 1f). This microtubule bundling activity of the ISB was independent of the dimerization domain, likely requiring

oligomerization through other regions (Supplementary Fig. 1F). We conclude that the CPC contains an additional microtubule-binding site on the N-terminus of the Borealin subunit.

**Borealin-MT interaction is important for faithful mitosis.** We generated stable HeLa TReX cells expressing N-terminal LAP (GFP + S-peptide)-tagged Borealin$^{WT}$, Borealin$^{MTBM}$ or Borealin$^{Δ20}$. We reduced the endogenous Borealin using a siRNA targeting the 3'UTR (Fig. 2a, c; Supplementary Fig. 1H) and imaged cells traversing mitosis by time-lapse microscopy with a cell permeable SiR-DNA dye. It took significantly longer for cells complemented with both Borealin$^{MTBM}$ and Borealin$^{Δ20}$ to

**Fig. 2** Interaction of both Borealin and INCENP with microtubules is important for error free mitosis. **a** Schematic of siBorealin-mediated knockdown rescue experiment. **b** Representative frames from time-lapse imaging of SiR-DNA labeled cells treated with Borealin siRNA and rescued with expression of indicated Borealin transgene, form one of three independent replicates. The quantitation is shown in **d** and **e**. **c** Western blots of cells expressing indicated transgene and treated as in **a** with Borealin or control siRNA, showing endogenous Borealin and LAP-Borealin expression. Tubulin staining is used as loading control. The lysates were collected 8–9 h after the second thymidine release. **d** Box and whisker graph of NEBD to anaphase duration from time-lapse movies. Statistical analysis was performed using one-way ANOVA (Kruskal–Wallis test) and Dunn's Multiple Comparison Test (for **b** and **d** data from three independent experiments $n = 233$ for LAP-Borealin$^{WT}$, $n = 204$ for LAP-Borealin$^{MTBM}$, and $n = 184$ for LAP-Borealin$^{\Delta 20}$). **e** Bar graph showing percent of cells undergoing anaphase with lagging chromosomes, cells were treated as described earlier in **a** and rescued with indicated Borealin transgene (data from $n = 3$ independent experiments, 63, 96, and 74 cells were analyzed for WT; 70, 84, and 50 cells were analyzed for MTBM; 68, 66, and 50 cells were analysed for Δ20). Open circle indicates the individual data points. Error bars are ±s.d. **f** Bar graph showing percent of cells undergoing anaphase with lagging chromosomes, treatment was done as in **a**, except during siRNA transfection step both Borealin and INCENP siRNA were added and the experiment was carried out in presence of 1 μg/ml Doxycycline to ensure INCENP transgene expression (data from $n = 2$ independent experiments, 110 and 91 cells were analyzed for INCENP$^{WT}$ Borealin$^{WT}$;59 and 100 cells were analyzed for INCENP$^{WT}$ Borealin$^{MTBM}$; 93 and 100 cells were analyzed for INCENP$^{\Delta SAH}$ Borealin$^{WT}$; 69 and 50 cells were analyzed for INCENP$^{\Delta SAH}$ Borealin$^{MTBM}$). Open circle indicates the individual data points. Error bars are ±s. d. Statistical analysis was performed using one-way ANOVA and Bonferroni's Multiple Comparison Test for both **d** and **e**. Bonferroni's Multiple Comparison Test was used for **f**. ***$P < 0.001$; **$P < 0.01$, *$P < 0.05$. All Box and whisker plots represent the median (central line), 25th–75th percentile (bounds of the box) and 5th–95th percentile (whiskers). Scale bar is 5 μm

---

traverse mitosis (NEBD to anaphase) than LAP-Borealin$^{WT}$ cells (Fig. 2b, d), demonstrating a function for the Borealin-microtubule interaction in early mitosis. Cells expressing MTBD mutants displayed a slight increase in duration from NEBD to metaphase, suggesting a minor role of CPC-microtubule interaction in chromosome congression (Supplementary Fig. 2A). We also assessed fidelity of chromosome segregation in absence of Borealin-microtubule interaction. Cell expressing either Borealin$^{MTBM}$ or Borealin$^{\Delta 20}$ had twice the frequency of anaphases with lagging chromatids than cells complemented with Borealin$^{WT}$ (Fig. 2e).

To directly test for a role in kinetochore-microtubule error correction, we first incubated cells with the Eg-5 kinesin inhibitor STLC to generate improper attachments, then washed the cells out of the drug and followed the fidelity of error correction by quantifying the number of cells in anaphase with lagging chromatids[43]. Cells expressing the Borealin$^{MTBM}$ or Borealin$^{\Delta 20}$ doubled the number of anaphases with lagging chromatids over controls demonstrating a requirement of the Borealin MBD in error correction (Supplementary Fig. 2C). We also replaced the N-terminal 20 amino acids of Borealin with a different microtubule-binding domain from PRC1 (Supplementary Fig. 2B). Cells expressing the chimeric protein resolved kinetochore-microtubule errors significantly better than the Borealin$^{\Delta 20}$ demonstrating that the key function of this domain is interaction with microtubules (Supplementary Fig. 2C). We conclude that Borealin microtubule-binding activity plays an important role in preventing and correcting improper kinetochore-microtubule attachments.

To understand the relative contributions of the two CPC microtubule-binding domains, we compared cells lacking the Borealin (Borealin$^{MTBM}$) or the INCENP (INCENP$^{\Delta SAH}$) MBD or both and determined the percent of cells undergoing anaphase with lagging chromosomes. Cells lacking either the Borealin or the INCENP MBD had two-fold more anaphases with lagging chromosomes (~40%) (Fig. 2f). Cells lacking both the MBDs had an even worse phenotype, with almost 75–80% cells undergoing anaphase with lagging chromosomes (Fig. 2f). These cells also showed an increase in duration of mitosis (Supplementary Fig. 2D). We conclude that the Borealin and the INCENP MBDs play different roles in the kinetochore-microtubule error correction process. Although, we note that the interpretation of this experiment is complicated by the fact that the SAH domain is also the region of the INCENP that is hypothesized to stretch in order to phosphorylate the kinetochore in the "dog leash" model[23].

We determined the duration of mitosis of Borealin depleted cells, which were complemented with Borealin$^{WT}$, Borealin$^{MTBM}$ or Borealin$^{\Delta 20}$ by live imaging upon treatment with 100 nM paclitaxel (Fig. 3a, b). Cells rescued with Borealin$^{MTBM}$ or Borealin$^{\Delta 20}$ arrested in mitosis for significantly shorter duration than the cells expressing Borealin$^{WT}$, in presence of taxol (Fig. 3c). We determined if replacing the MBD of Borealin with the MBD of PRC1 could rescue the defective SAC arrest (Supplementary Fig. 2B). We observed a partial rescue in the duration of taxol-induced SAC arrest in the cells expressing the PRC1$^{MBD}$-Borealin$^{\Delta 20}$ chimeric protein compared to the microtubule-binding mutants of Borealin (Fig. 3c, d). Increasing the amount of Aurora-B at centromeres by expressing CENPB-INCENP$^{747-918}$ (Supplementary Fig. 2F) did not rescue the SAC defect (Fig. 3e), further arguing that the central defect in the Borealin$^{MTBM}$ is microtubule binding and not the partial reduction in centromeric CPC, which we describe below. We conclude that the reason for deficiency in SAC maintenance in the Borealin mutants is the inability of CPC to bind microtubules.

**MTs adjacent to the inner centromere stimulate CPC activity.** Next, we sought to understand a physiological role for the binding of the CPC to microtubules, which is known to lead to its activation. We hypothesized that microtubules that run adjacent to the inner centromere and extend to the kinetochores, would stimulate phosphorylation of kinetochore substrates. This hypothesis is consistent with microtubule configurations observed in prometaphase and at merotelics, and would explain why such attachments do not persist. It is, however, unclear how cells could obtain stable correct attachments, if end-on microtubules also stimulated CPC activation. Thus, we turned to quantitative theoretical investigation to explore the consequence of microtubule stimulation of CPC activity.

We extended a previously developed mathematical model of spatial regulation of CPC activity built upon the fact that Aurora-B kinase can activate itself and is inactivated by a phosphatase (Fig. 4a, upper box). In the one-dimensional model of spatial Aurora-B regulation, these non-linear reactions engage the chromatin-bound and soluble CPC pools, which together with a soluble phosphatase form a bi-stable spatially distributed reaction-diffusion system[24]. At an unstretched centromere (as in metaphase), the concentration of chromatin-bound CPC exceeds activation threshold, so kinase activity is high at the Ndc80 site of the kinetochore (Fig. 4b). However, when chromatin is stretched, as in metaphase, the kinase activity falls

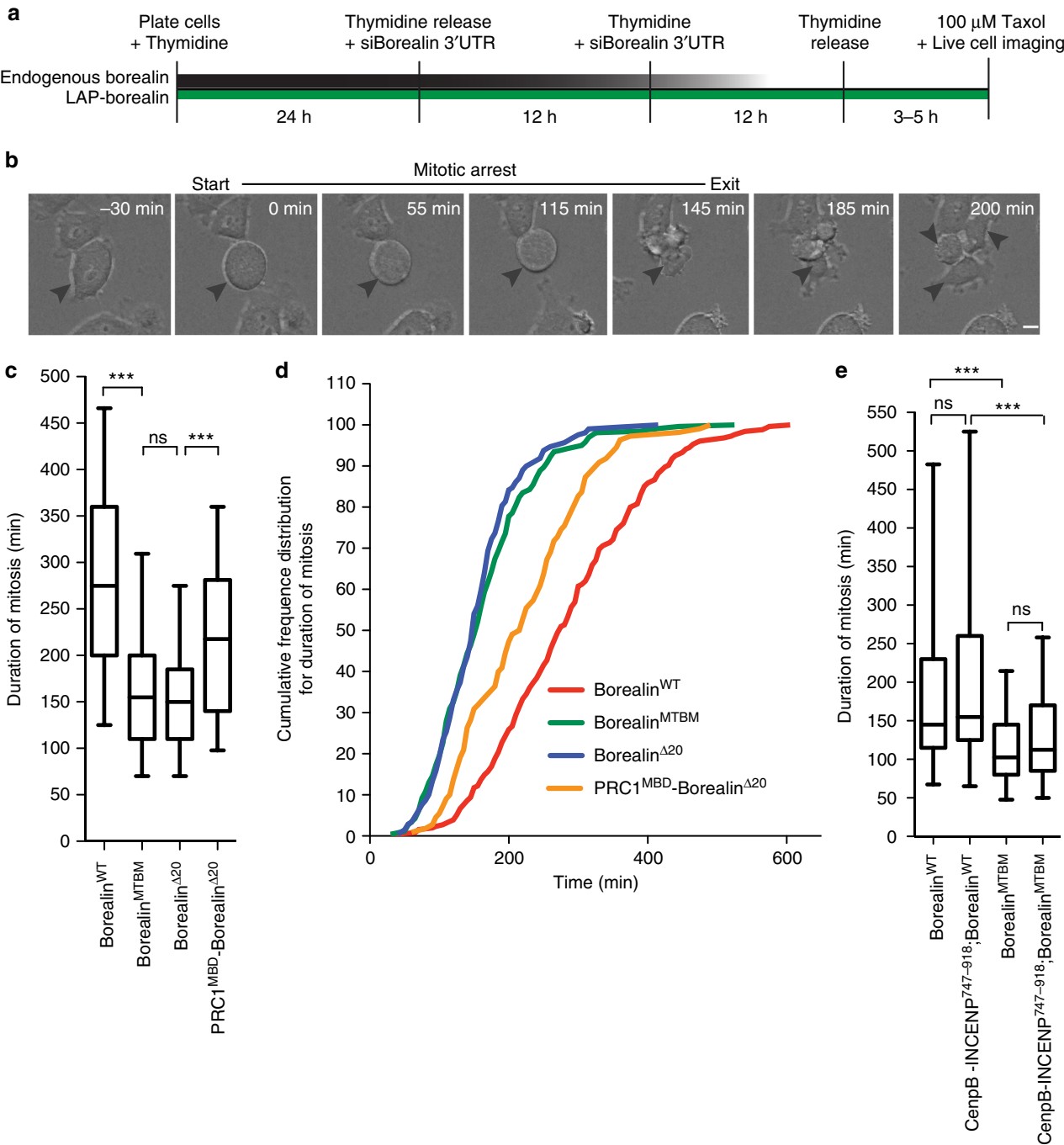

**Fig. 3** CPC-microtubule interaction is important for maintenance of taxol dependent spindle assembly checkpoint arrest. **a** Schematic of experimental procedure for **b–d**. **b** Representative time-lapse phase-contrast images, from one of two independent replicates, of a cell treated as in **a**. Arrow head points to the cell that enters mitosis and is arrested in presence of 100 nM taxol and exits mitosis, time of mitotic entry and exit are depicted. Duration of mitotic arrest is the duration between mitotic entry and exit. **c** Box and whisker graph of duration of mitosis in cells expressing the indicated Borealin transgene ($n = 255$ for Borealin[WT], $n = 261$ for Borealin[MTBM], $n = 209$ for Borealin[Δ20], and $n = 110$ for PRC1[MBD]-Borealin[Δ20], combined data from two independent repeats). **d** Graph of data from **c** showing cumulative frequency distribution of duration of mitosis for the cells expressing indicated Borealin transgene. **e** Box and whisker graph of duration of mitosis in presence of 100 nM taxol. Cells were treated as in **a** with the exception that 1 μg/ml of doxycycline was added, at the time of second thymidine addition, for induction of CenpB[DBD]-INCENP[747-918] expression ($n = 109$ for Borealin[WT] and CenpB[DBD]-INCENP[747-918] Borealin[WT], $n = 110$ for Borealin[MTBM], and $n = 106$ for CenpB[DBD]-INCENP[747-918] Borealin[MTBM], combined data from two independent repeats). Statistics performed using Dunn's Multiple Comparison Test, ***$P < 0.001$; **$P < 0.01$, *$P < 0.05$ and ns $P > 0.05$. All Box and whisker plots represent the median (central line), 25th–75th percentile (bounds of the box) and 5th–95th percentile (whiskers). Scale bar is 5 μm

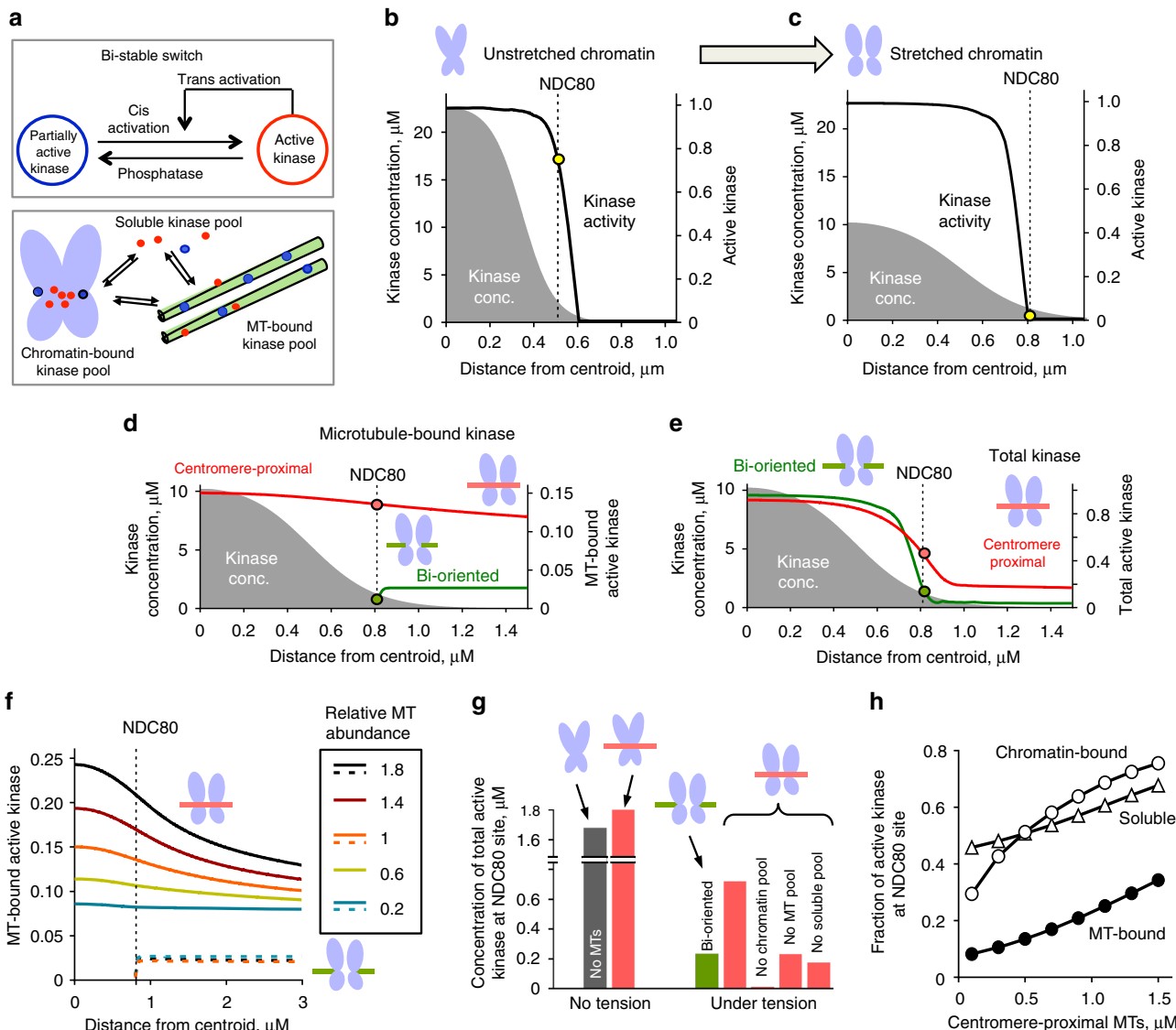

**Fig. 4** Mathematical model of tension-dependent and microtubule configuration-dependent phosphoregulation at kinetochores. **a** Upper box: schematic for kinase-phosphatase switch. Lower box: schematic for dynamically exchanging CPC pools. Kinase within each pool becomes activated by auto-phosphorylation and inactivated by soluble phosphatase, see Methods for details. Unless stated otherwise, calculations for all panels were done using model parameters listed in Supplementary Table 6 for different centromeric tension and/or microtubule configurations. **b**, **c** The concentration of chromatin-bound CPC kinase (gray, left axis) decreases from the middle of the centromere (centroid) to outer kinetochore containing Ndc80 substrate (shown with broken line). Because the total amount of chromatin-bound kinase does not change under tension, when chromatin is stretched, the kinase concentration is reduced throughout the centromere. Black lines (right axes) show the fraction of chromatin-bound kinase that is active; these curves drop abruptly due to bi-stability. Yellow dots show activity of chromatin-bound kinases at the Ndc80 location. These panels show tension-dependent model behavior in the absence of the microtubule-bound kinase pool ($k_{on}^{MT} = 0$). **d**, **e** Profiles as in **c** for different microtubule configurations: bi-oriented (green) or with centromere-proximal microtubules (red). Left axes—concentration of chromatin-bound kinase under tension (gray). Colored curves are plotted using the right axes, showing fraction of active kinase for all CPC pools (**e**) or only for the microtubule-bound pool (**d**). Colored dots show fraction of active kinase at Ndc80 site; the corresponding concentrations for dots on panel **d** are plotted in panel **g** "under tension". **f** Spatial distribution of active microtubule-bound kinase (fraction from the microtubule-bound kinase pool) for different microtubule abundance (parameter $\alpha$), see panel **d** for other details. Solid and dashed lines correspond to configurations with centromere-proximal and end-on microtubules, respectively. **g** Calculated concentration of active kinase (sum of all pools) at Ndc80 location of the kinetochore for indicated configurations. The last three columns correspond to model predictions in the presence of tension and centromere-proximal microtubules but either chromatin, microtubule or soluble CPC pools are eliminated. **h** Fractions of active kinase in all pools—microtubule-bound, chromatin-bound and soluble—respond to the size of centromere proximal microtubule bundle

below the activation threshold before it reaches the Ndc80 site (Fig. 4c). Here we incorporated the ability of the CPC to bind and diffuse along the microtubules in to this model (Fig. 1c, Supplementary Fig. 1C–E). In this improved model, three CPC pools (microtubule-bound, chromatin-bound and soluble) exchange quickly (Fig. 4a lower box), while simultaneously

engaging in phosphorylation-dephosphorylation reactions. Detailed description of the model, its assumptions and parameter values are provided in Methods.

Our model demonstrates that the CPC has low activity on microtubules that are located away from the centromere (Fig. 4d, green curve), implying that CPC binding to the end-on

microtubules of the stretched kinetochores does not increase Ndc80 phosphorylation (Fig. 4e, green curve). Importantly, the fraction of the active microtubule-bound kinase is much higher on centromere-proximal microtubules for the same model parameters (Fig. 4d, red curves). A gradient of active kinase forms along these microtubules with its maximum coinciding with a peak of the chromatin-bound CPC and decreasing toward the kinetochore. We varied the abundance of the centromere-proximal microtubules, and microtubule-associated kinase activity at the Ndc80 site increased with more microtubules (Fig. 4f). The exact gradient of the microtubule-bound CPC as a function of the number of attached microtubules could not be calculated because little is known about parameters of CPC-microtubule interactions in cells. However, same concentration of microtubules led to either low or high kinase activity at the Ndc80 site, depending on whether these microtubules were in the end-on configuration or centromere-proximal (Fig. 4f). Thus, the model predicts that different microtubule configurations can dictate distinct kinase activity outputs at kinetochores in this complex kinase-phosphatase system.

We calculated total kinase activity at the Ndc80 site resulting from the combined activities of the microtubule-bound, chromatin-bound and soluble kinase pools to further understand the mechanism of this phosphoregulation. Centromere-proximal microtubules generated an expanded phosphorylation zone relative to the end-on configuration (Fig. 4e red vs. green curves), which increased the total kinase activity at the Ndc80 site (Fig. 4g green vs. neighboring red column). This is remarkable because in these different microtubule configurations the kinetochores were similarly stretched. Thus, microtubule binding by the CPC can override the tension-dependent regulation in the presence of centromere-proximal microtubules.

To test whether the expansion of the kinase activity gradient along the centromere-proximal microtubules was indeed induced by their proximity to the centromere, we eliminated chromatin-bound CPC pool in the model by setting the association constant to chromatin-binding sites to zero. This completely prevented kinase activity at the Ndc80 site (Fig. 4g, Supplementary Fig. 3A), demonstrating that a centromere-bound kinase pool activates the microtubule-bound kinase. Preventing the CPC-microtubule interaction reduced kinase activity to the same level as at the bi-oriented kinetochores (Fig. 4f). Furthermore, depletion of the soluble CPC pools had negative impact, implying that the effects from all CPC pools were synergistic. Consistently, we found that each of the kinase pools, not just the microtubule-bound one, was responsive to the abundance of centromere-proximal microtubules (Fig. 4h). Together, these findings strongly suggest that microtubule-binding affinity of the CPC can control kinetochore phosphorylation in response to the abundance of centromere-proximal microtubules.

**Borealin-MT interaction enables kinetochore phosphorylation.** A major prediction of the modeling study is that robust phosphorylation of kinetochores, in presence of centromere-proximal microtubules, depends on the microtubule-bound CPC. To test whether Borealin-microtubule binding is required for increased phosphorylation of kinetochores, we measured kinetochore phosphorylation by the CPC in prometaphase cells expressing the Borealin$^{WT}$ or Borealin$^{MTBM}$ (Fig. 5a) using a series of phospho-antibodies by quantitative immunofluorescence[15,17]. Phosphorylation of a number of CPC substrates at the kinetochore (DSN1 pS109; KNL1 pS60, CenpA pS7 and Hec1 pS44 of the Ndc80 complex) was reduced when Borealin-microtubule interaction was disrupted (Fig. 5b–e, Supplementary Fig. 4A–F). In contrast, the chromatin substrate H3 pS10 was not affected (Fig. 5f, g). The

reduction in the CPC phosphorylation was not due to defective kinetochore assembly (Supplementary Fig. 4G–J). Interestingly, we also saw that a recently reported Aurora-A substrate at the kinetochore, Hec1 pS69[44], was also reduced in the Borealin$^{MTBM}$ compared to the Borealin$^{WT}$ expressing cells (Fig. 5h, i). Aurora-A centromere localization depends on its interaction with INCENP, which is important for Hec1pS69 phosphorylation[44], and thus might also depend on Borealin. Phosphorylation by the CPC on DSN1 pS109 was partially rescued by replacing Borealin MBD with PRC1 MBD, which is consistent with our observation that an exogenous MBD can rescue SAC and error-correction functions (Supplementary Fig. 4E, F). We conclude that robust kinetochore phosphorylation depends upon the ability of borealin to bind microtubules.

**Borealin-MT interaction enhances inner-centromere localization.** The interaction of Borealin with microtubules could regulate the amount of the CPC bound to the centromere, the transfer of the signal from the inner centromere to kinetochores or both of these steps. Note that the former is a complexity that was not included in our model and confounds the interpretation of the above finding about Borealin$^{MTBM}$. To examine this possibility we measured the levels of the centromeric CPC in cells expressing the Borealin$^{MTBM}$. The centromeric CPC levels were reduced in the Borealin microtubule-binding mutants compared to wild type, as assessed by the immunostaining for Aurora-B, INCENP and Borealin (Fig. 6a–d). The reduction in the CPC localization is not due the N-terminal LAP-tagging of the Borealin protein as C-terminally tagged Borealin$^{MTBM}$ also shows the same reduction in inner-centromeric CPC localization (Supplementary Fig. 5A, B). Defective centromeric CPC localization is not due to defective dimerization by Borealin, which was shown to regulate CPC localization[45], since disabling MTBD and dimerization domain led to additive reduction in CPC localization, confirming that these are separable functions on the Borealin protein (Supplementary Fig. 5C, D). The reduction in level of activated Aurora-B, measured by T-loop phosphorylation, is comparable to the reduction in total CPC levels in the inner-centromere (Supplementary Fig. 6A, B), indicating that the clustering dependent activation of the inner-centromeric CPC is not affected in cells expressing the Borealin$^{MTBM}$. Combining the Borealin$^{MTBM}$ with deletion of INCENP's SAH microtubule binding domain reduced the amount of CPC (~10%) compared to either mutant alone (Supplementary Fig. 2E).

We tested whether the Borealin MBD stimulates CPC localization at the centromere indirectly by enabling the CPC to interact with centromere proximal microtubules. In wild-type cells, low doses of nocodazole allow short microtubules to remain around centromeres and stimulate the localization of the CPC to the inner centromere[35]. The stimulation of CPC localization was significantly attenuated in cells expressing the Borealin$^{MTBM}$ at low nocodazole concentration (0.33μM) (Fig. 6e, f). In contrast, the amount of inner-centromeric CPC was similar in mutant and wild-type cells in presence of 3.3 μM nocodazole (high concentration), which completely eliminates centromere proximal microtubules (Fig. 6e, f). These data suggest that the Borealin MBD interacts with centromere proximal microtubule that stimulates CPC activity at kinetochores and hence localization to the inner-centromere. Moreover, the fact that the CPC levels are similar in high doses of nocodazole argues that the chromatin-binding properties of the CPC are not affected in Borealin$^{MTBM}$.

To gain insight into molecular mechanism of the CPC localization defect, we examined whether the Borealin MBD controls either of the two-histone phosphorylation feedback loops

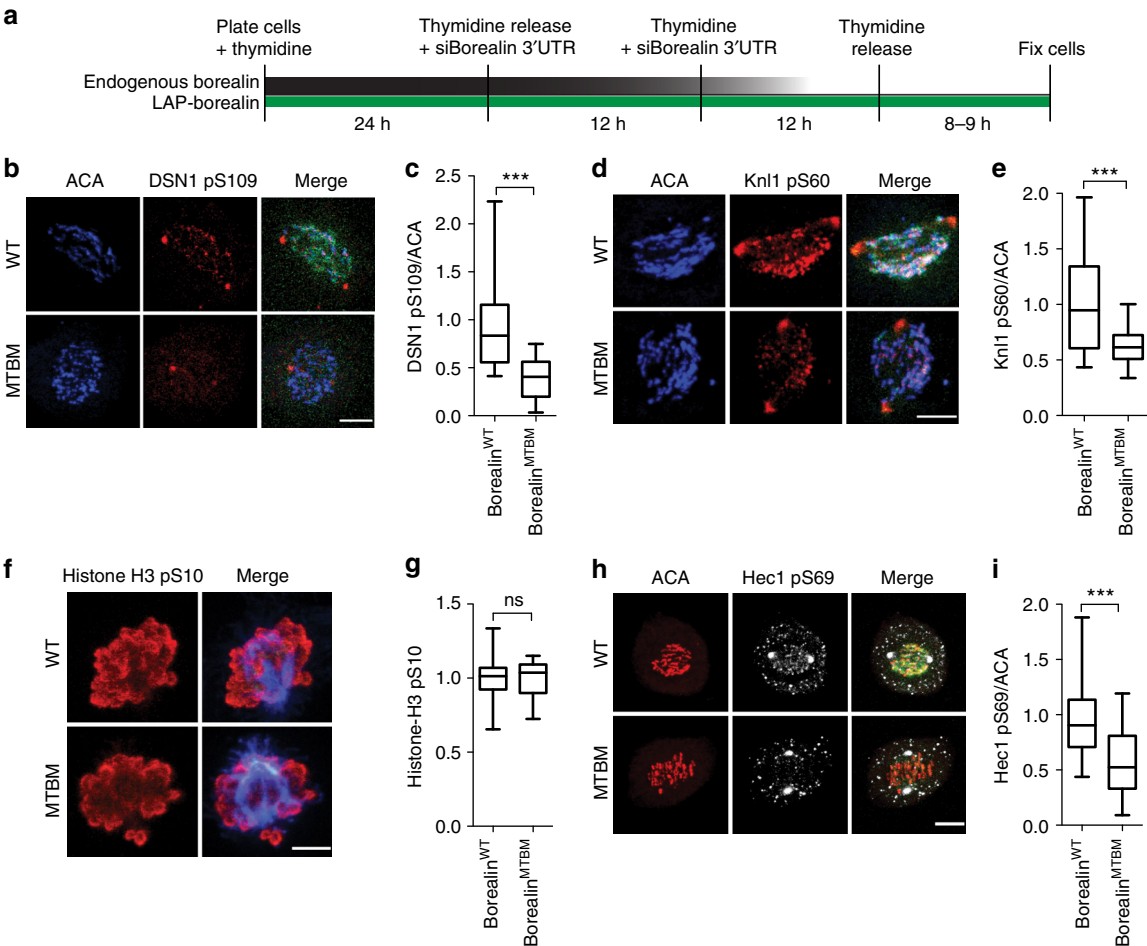

**Fig. 5** Borealin-microtubule interaction contributes to robust phosphorylation of the kinetochore substrates by the CPC. **a** Schematic of Borealin siRNA mediated knockdown rescue experiment. HeLa-TReX cells stably expressing LAP-Borealin[WT or MTBM] were treated with Borealin 3'UTR siRNA and cells were immunostained in first mitosis after knockdown of endogenous Borealin. Kinetochore phosphorylation was assessed by immunostaining with **b** DSN1 pS109, **d** Knl1 pS60, **h** Hec1 pS69, antibodies. Green color in merge images indicates LAP-Borealin localization. **f** Chromatin phosphorylation was assessed by immunostaining with histone H3 pS10. Box and whisker graphs of normalized intensity from **c** DSN1 pS109 ($n = 103$ (from 8 cells) for WT and $n = 73$ (from 9 cells) for MTBM), **e** Knl1 pS60 ($n = 61$ for WT and $n = 72$ for MTBM, from eight cells per condition), **g** H3 pS10 ($n=18$ for WT and $n=11$ for MTBM) and **i** Hec1 pS69 ($n = 68$ (from 7 cells) for WT and $n = 73$ (from 6 cells) for MTBM, from at least six cells per condition), staining. All representative immunofluorescence images and quantitation data are from 1 of 2 independent repeats. Statistical analysis was performed using Mann Whitney Test, ***$P < 0.0001$ and ns $P > 0.05$. All Box and whisker plots represent the median (central line), 25th–75th percentile (bounds of the box) and 5th–95th percentile (whiskers). Scale bar is 5 μm

that localize the centromeric CPC. Both the H2A pT120 phospho-histone mark and the Sgo1 levels at the kinetochores are both reduced about 25% in the microtubule-binding mutants (Fig. 7b, c; Supplementary Fig. 6C, D). However, the Haspin-dependent Histone H3pT3 phospho-mark is unchanged in the Borealin WT and the MTBM expressing cells (Fig. 7d, e). We conclude that Borealin MBD stimulates the kinetochore axis of the CPC localization pathways. This result is consistent with reduced kinetochore phosphorylation in cells expressing the Borealin[MTBM], since the CPC controls the H2ApT120 pathway by phosphorylation of kinetochore substrates MPS1 and the Ndc80 complex[46–49].

**MT bound non-centromeric CPC phosphorylates kinetochores**. We next sought to test directly whether the Borealin microtubule interaction was controlling the transfer of the CPC activity from the inner centromere to kinetochores, as predicted by the model. We first tested whether the non-centromeric pool of the CPC contributes to kinetochore phosphorylation in human cells by developing an assay to manipulate the non-centromeric CPC

without affecting the amount of Aurora-B at centromeres. Specifically, we artificially targeted the Aurora-B kinase to the centromere by fusing Aurora-B binding domain of INCENP, INCENP[747-918] with the DNA binding domain of the CENP-B, which recognizes alpha satellite DNA. This chimeric protein should be incapable of stretching significant distances and binding microtubules since it cannot bind Borealin and Survivin and lacks the SAH domain of INCENP (Fig. 8b). We inhibited the targeting of the endogenous CPC to the inner-centromere by adding the Haspin inhibitor (5-ITU) and depleting Bub1 (Supplementary Fig. 7A–E) and measured Aurora-B kinase activity at the outer kinetochore using two different antibodies raised against Hec1-pS44 and Hec1-pS55. Under these conditions, Hec1 was still phosphorylated at the same level as in unperturbed cells (Supplementary Fig. 7F, G). This is consistent with recent reports[37,50] and argues that the elongated stretch of INCENP cannot be the only mode of kinetochore phosphorylation (Fig. 8a, c, d; Supplementary Fig. 7F, G). These results support the reaction-diffusion model, in which combination of the diffusible and chromatin-bound CPC pools coordinates the

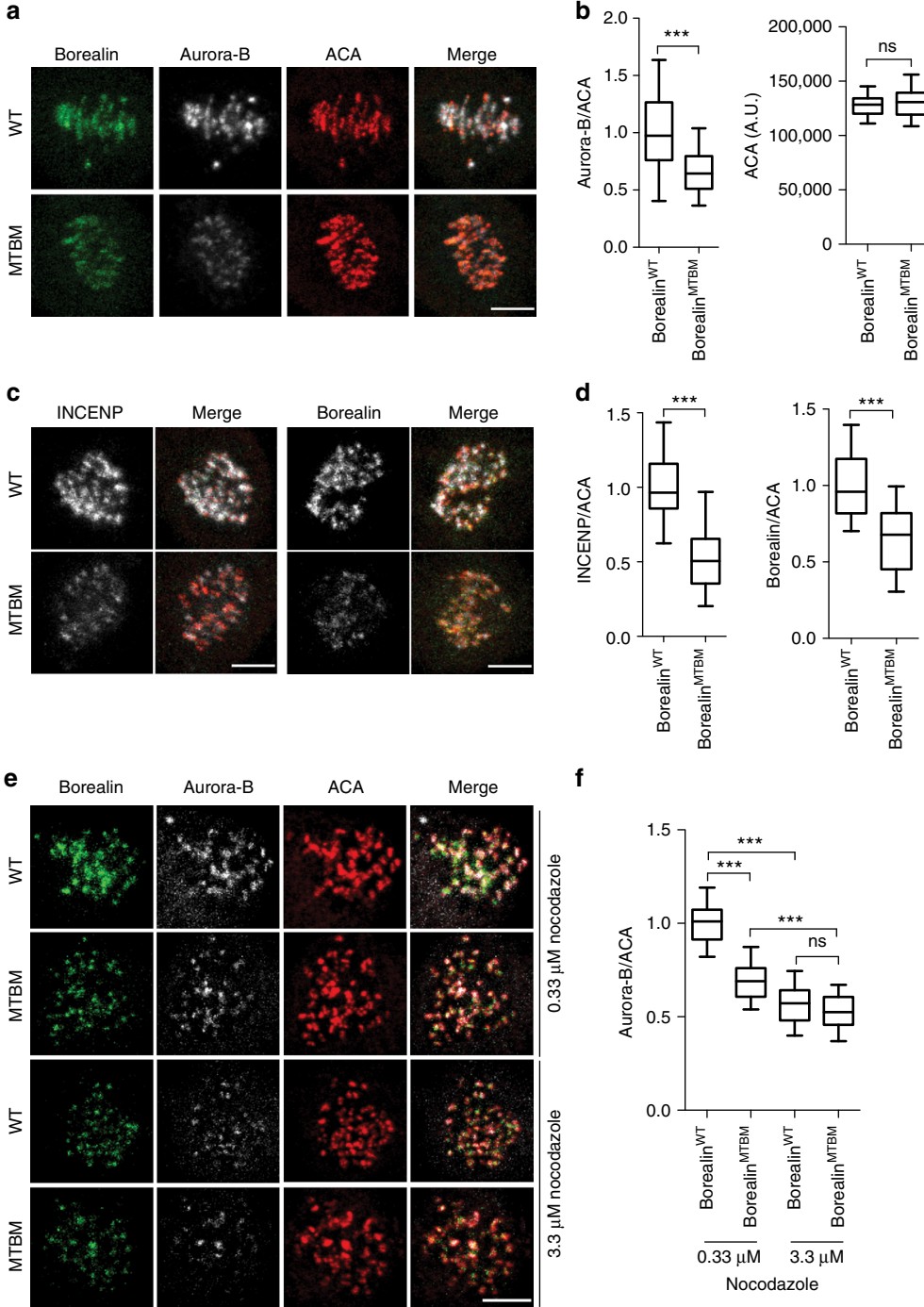

**Fig. 6** Borealin-microtubule interaction drives microtubule-dependent enhancement of the CPC localization to the inner-centromere. HeLa-TReX cells expressing LAP-Borealin[WT or MTBM] were treated with siRNA as described in Fig. 2a, and immunostained with ACA and Aurora-B (**a**) or Borealin (**c**) or INCENP (**c**) antibodies. **b** Box and whisker plots of Aurora-B intensity normalized to ACA ($P < 0.0001$) and absolute ACA intensity ($P = 0.1396$), Unpaired $T$-test with Welch's correction was applied ($n = 94$ for WT and $n = 110$ for MTBM) (representative images in **a** and quantitation in **b** from 9 cells per condition, from one of three independent replicates). **d** Box and whisker plots of INCENP ($P < 0.0001$) ($n = 84$ for WT and $n = 110$ for MTBM) and Borealin ($P < 0.0001$) ($n = 78$ for WT and $n = 76$ for MTBM) intensity normalized to ACA (representative images in **c** and quantitation in **d** from eight cells per condition from one of two independent replicates). Statistical analysis was performed using Mann Whitney Test. **e** HeLa-TReX cells expressing LAP-Borealin[WT or MTBM] were treated as described in Fig. 2a, and incubated with either 0.33 μM or 3.3 μM nocodazole for 45 min followed by immunostaining with ACA and Aurora-B. **f** Box and whisker of Aurora-B intensity normalized to ACA ($n = 85$ for WT and $n = 167$ for MTBM in 0.33 nocodazole) ($n = 140$ for WT and $n = 100$ for MTBM in 3.3 μM Nocodazole) (representative images in **e** and quantitation data in **f** from one of two independent replicates with eight cells analyzed per condition). Statistics performed using Dunn's Multiple Comparison Test, ***$P < 0.001$; **$P < 0.01$, *$P < 0.05$ and ns $P > 0.05$. All Box and whisker plots represent the median (central line), 25th–75th percentile (bounds of the box) and 5th–95th percentile (whiskers). Scale bar is 5 μm

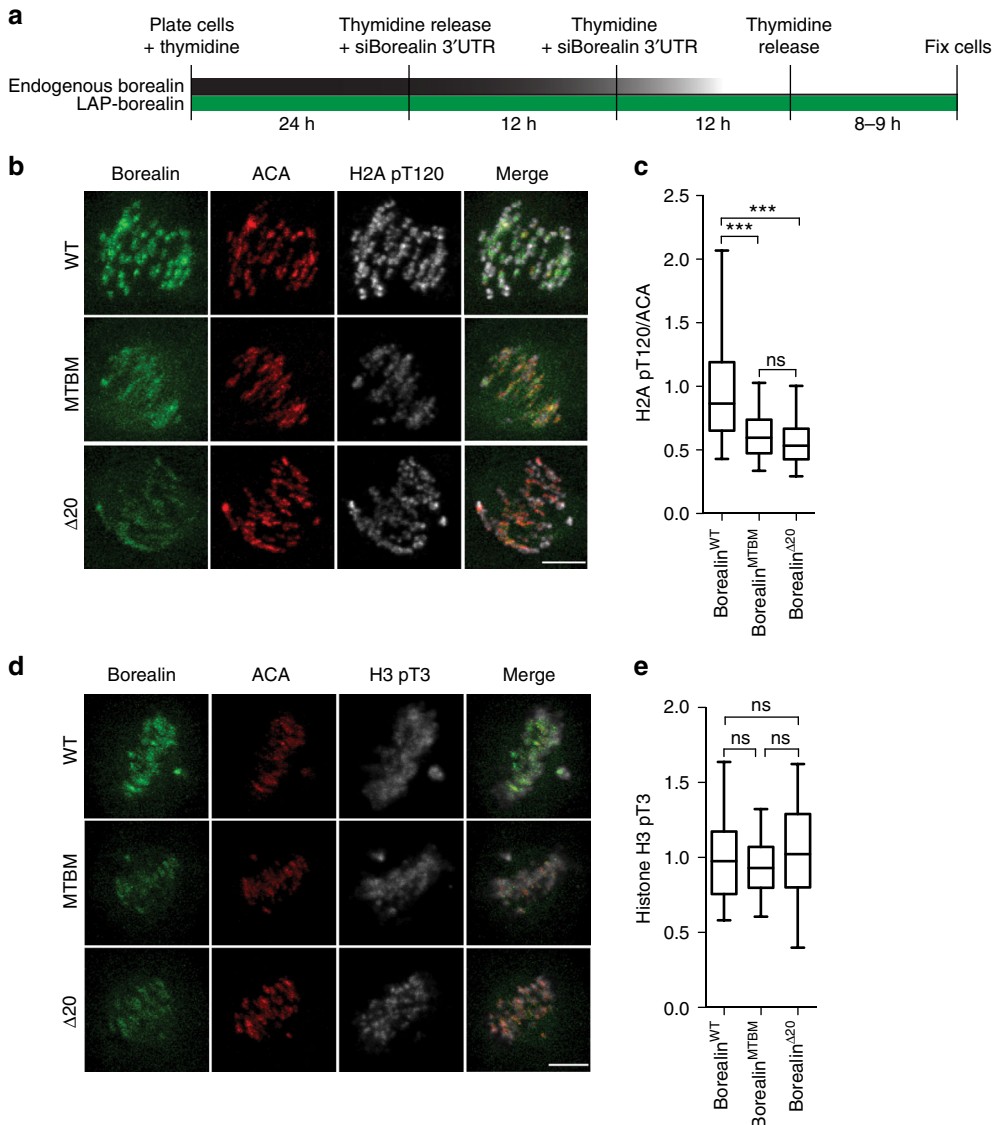

**Fig. 7** Borealin-microtubule interaction enhances kinetochore sub-network of the CPC inner-centromere localization pathway. (**a**) Schematic of Borealin siRNA mediated knockdown rescue experiment. (**b**, **d**) HeLa-TReX cells stably expressing LAP-Borealin[WT or MTBM or Δ20] were treated as in **a** and immunostained with histone H2a pT120 and H3 pT3 antibodies. **c** Box and whisker graph of histone H2a pT120 intensity normalized by ACA intensity ($n =$ 166 (from 13 cells) for WT, $n = 223$ (from 16 cells) for MTBM, and $n = 150$ for Δ20 (from12 cells)). **d** Box and whisker graph of histone H3 pT3 intensity ($n = 188$ (from 16 cells) for WT, $n = 200$ (from 15 cells) for MTBM, and $n = 100$ (from 10 cells) for Δ20). Statistical analysis was performed using one-way ANOVA (Kruskal–Wallis test) with Dunn's Multiple Comparison Test, ***$P < 0.001$; **$P < 0.01$, *$P < 0.05$ and ns $P > 0.05$. All representative images and quantitation are from one of two independent replicates. All Box and whisker plots represent the median (central line), 25th–75th percentile (bounds of the box) and 5th–95th percentile (whiskers). Scale bar is 5 μm

phosphorylation of the kinetochore[24,25,31]. We specifically depleted the non-centromere pool of the CPC by depleting the Borealin or INCENP subunit in cells expressing CENP-B-INCENP[747-918]. Strikingly, depleting Borealin or INCENP in these cells reduced the Hec1 phosphorylation compared to control cells, even though the amount of Aurora-B at the centromere was unaffected (Fig. 8a, c, d; Supplementary Fig. 7H–L). Since the CENP-B DNA binding domain that is used to target Aurora-B is known to turnover[25], it likely has a non-centromeric pool of its own. Therefore, the requirement of Borealin suggests that the non-centromeric pool of the CPC must use some activity that is lacking in the CENP-B-INCENP[747-918] to phosphorylate kinetochores.

We hypothesized that the diffusible pool of the CPC is activated by the inner centromere pool and then uses

microtubules to travel to the kinetochore to phosphorylate kinetochore substrates, as predicted by the model and single molecule binding assays. As an initial test of this idea we measured the requirement for Borealin-microtubule interaction in our system that isolated the requirement for soluble CPC to phosphorylate kinetochores. Specifically, we compared Hec1 phosphorylation in cells rescued with Borealin[WT] or Borealin[MTBM] and expressing CENP-B-INCENP[747-918] upon depletion of the endogenous CPC localization pathways (Fig. 8e–i, Supplementary Fig. 7M–P). We found that even though active Aurora-B was targeted to the inner centromere, robust phosphorylation of kinetochores still required the Borealin MBD on the non-centromeric pool of the CPC (Fig. 8e–i; Supplementary Fig. 7M–P). In conclusion, we have identified two roles for microtubule binding in Borealin activity. First, the Borealin MBD

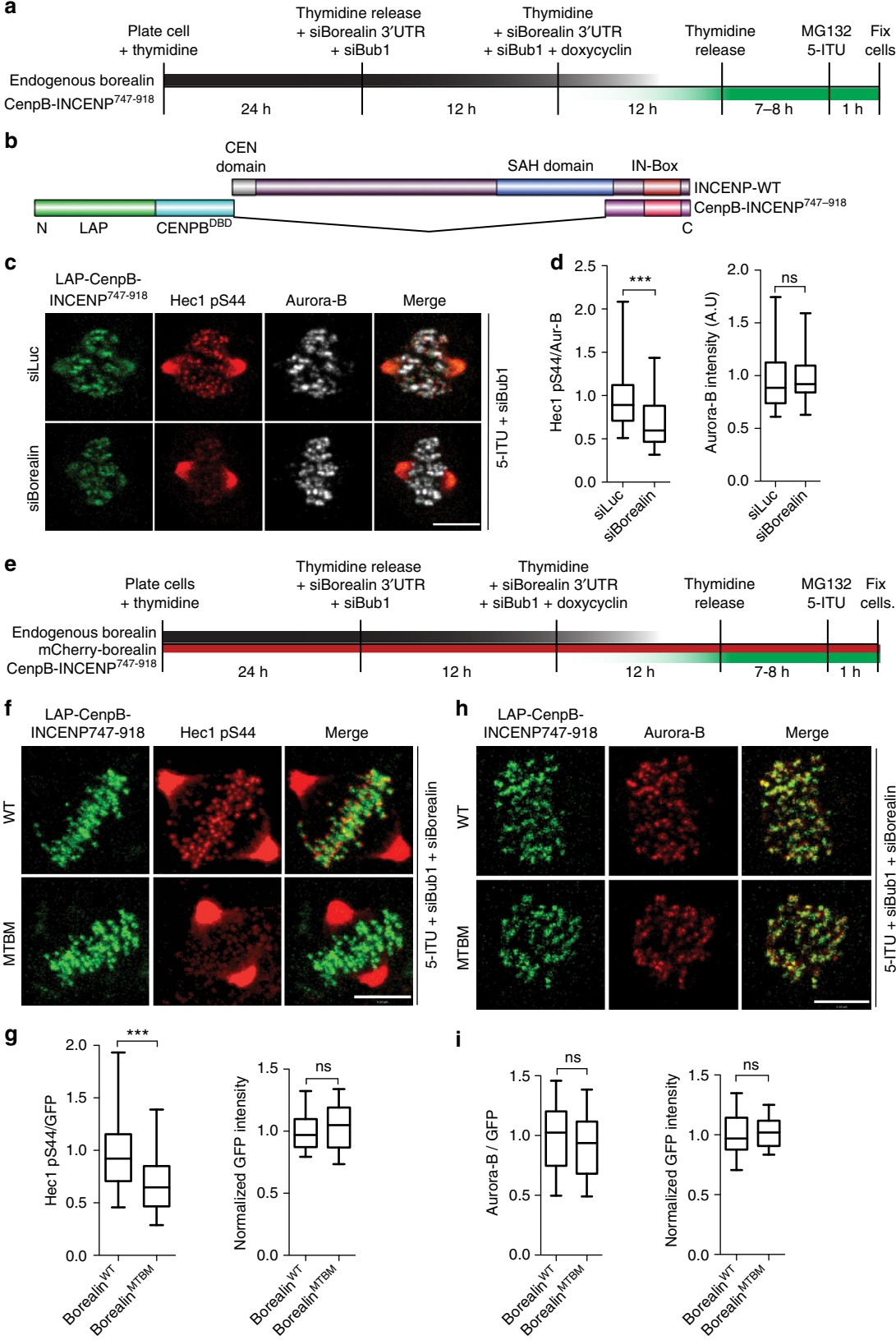

allows the activated non-centromeric CPC to phosphorylate kinetochore substrates. Second, this kinetochore phosphorylation affects the Bub1-Sgo1-CPC localization pathway increasing the amount of CPC in the inner centromere.

## Discussion

Here we provide evidence that a microtubule-binding site on the Borealin subunit of the CPC is required for its mitotic functions. The MTBD resides in the N-terminus of Borealin near the triple

**Fig. 8** Borealin-mediated non-centromeric CPC-microtubule interaction is required for robust phosphorylation of the kinetochore substrates by the CPC. **a** Schematic of experimental procedure, for **c** and **d**, Borealin and Bub1 siRNA-mediated knockdown of endogenous Borealin and Bub1 in cells expressing chimeric CenpB[DBD]-INCENP[747-918] transgene. **b** Cartoon showing domain structure of INCENP[WT] and chimeric LAP-CenpB-INCENP[747-918] protein. **c** Cells expressing LAP-CenpB- INCENP[747-918], for targeting Aurora-B to the centromeres, were treated with control siRNA (siLuc) or siBorealin to deplete endogenous Borealin as shown in **a**. Bub1 siRNA and Haspin inhibitor, 2 μM 5-ITU, were added before fixation in order to remove the inner-centromeric CPC localization signal. Cells were immunostained with antibodies against Hec1 pS44 and Aurora-B and representative images are shown. **d** Box and whisker graph of normalized Hec1 pS44 intensity and normalized Aurora-B intensity ($n = 150$ (from 13 cells) for cells treated with siLuc and $n = 117$ (from 12 cells) for cells treated with siBub1 and siBor). **e** Schematic of experimental procedure for **f–i**. **f** Cells were treated as in **e**, LAP-CenpB- INCENP[747-918] and LAP-Borealin[WT] or LAP-Borealin[MTBM] expressing cells were treated with siBub1 and haspin inhibitor (2 μM 5-ITU) to delocalize the endogenous CPC from the inner-centromere. Endogenous Borealin was depleted with siBorealin treatment and cells were immunostained with Hec1 pS44 (**f**) and Aurora-B (**h**) antibodies, representative images are shown. **g** Box and whisker graph of normalized Hec1 pS44 and normalized GFP intensity from **f** ($n = 186$ (from 12 cells) for WT and $n = 171$ (from 8 cells) for MTBM). **i** Box and whisker graph of normalized Aurora-B and GFP intensity from H ($n = 135$ for WT and $n = 147$ for MTBM; data from 11 cells per condition). Statistical analysis performed using Mann Whitney Test, ***$P < 0.0001$ and ns $P > 0.05$. All representative immunofluorescence images and quantitation is from one of two independent replicates. All Box and whisker plots represent the median (central line), 25th–75th percentile (bounds of the box) and 5th–95th percentile (whiskers). Scale bar 5 μm

helix that generates the interface with INCENP and Survivin. Positively charged amino acids on Borealin mediate microtubule binding suggesting that the interaction is electrostatic in nature. INCENP may also contribute to a composite microtubule-binding surface though additional basic residues in this region (R43, R47).

The ISB MTBD described here is the third direct microtubule-binding region on the CPC along with the PR/SAH domain on INCENP and a microtubule binding region on Aurora-B/IN-Box[29,35–37,40]. Simultaneous depletion of microtubule binding activities on INCENP and borealin results in an increase in cells with anaphase lagging chromosomes compared to removing either alone. The CPC also binds other microtubule interacting proteins like kinesin MKLP-2, EB1, and GTSE1, suggesting that indirect interactions may also regulate the CPC[35,51–54]. Microtubules regulate Aurora kinase activity but it is unclear if there are additional reasons why the CPC requires these multiple microtubule interactions. For example, the CPC may play a structural role in building microtubule structures. Consistent with this idea the ISB complex built microtubule bundles in vitro and the CPC is intimately tied to microtubule bundle structures in vivo, as it both localizes to and coordinates the assembly of midzone microtubules in anaphase and preformed K-fibers in prometaphase cells[6,35,55–57]. In fact, a similar mutant in the borealin MTBD has been shown to have reduced localization to midzone microtubule bundles[42].

The MTBD of Borealin is important for full kinetochore phosphorylation and mitotic functions. This finding explains recent results demonstrating that the centromere-targeting domain of the CPC has roles independent of centromere targeting during paclitaxel-dependent checkpoint arrest[37], and why multiple-microtubule binding domains that are required for viability in yeast[28,29]. How the CPC in the inner centromere controls phosphorylation of adjacent kinetochores is an important unanswered question. We provide mathematical modeling and in vivo evidence that the inner centromere pool of the CPC is required to generate active kinase, but the spreading of kinase activity to kinetochores is enhanced by its ability to interact with microtubules. Moreover, we show that the CPC that phosphorylates kinetochores needs to interact with microtubule but not chromatin.

We propose that microtubule-dependent spreading of the CPC during prometaphase enables robust kinetochore phosphorylation independent of tension until the obtainment of end-on attachments, and that during metaphase it mediates selective detachment of the merotelic microtubules. Most models have suggested that the phosphorylation of kinetochores by the CPC is reduced after kinetochores obtain bipolar microtubule attachments because the ensuing pulling force increases the physical distance between the inner centromere and substrates (tension). Here we provide evidence that

the CPC can also regulate kinetochores by a "tension-independent" mechanism, where microtubules adjacent to inner-centromeres enable robust phosphorylation even if kinetochores are fully stretched. Since CPC binding to microtubules stimulates its kinase activity in vitro, the idea that specific microtubule structures stimulate CPC phosphorylation of kinetochores is appealing. There are at least three chromosome configurations on the mitotic spindle that would place inner centromeres adjacent to microtubules to stimulate robust kinetochore phosphorylation. First, they may be generated by preformed K-fibers, which are microtubule bundles that specifically bind the CPC and emanate from kinetochores before they generate end-on microtubule attachments[35,58,59]. Second, merotelic attachments bring the merotelic K-fiber microtubules adjacent to inner centromeres[2,3,60]. Third, electron micrographs have suggested that the microtubule density in the center of the prometaphase spindle excludes chromosomes so that the chromosome arms fold back so that telomeres are pointed away from the spindle and inner centromeres are proximal to the spindle microtubules[33] (Fig. 9a). In these cells many kinetochores obtain full interkinetochore stretch using lateral kinetochore attachments, presumably through the CENP-E motor[33]. Since this configuration also places inner centromeres adjacent to the central spindle microtubules, our model can explain how kinetochores remain in a high phosphorylation state independent of the increased distances generated by spindle pulling forces. We suggest that maturation of these initial microtubule attachments into "end-on" attached K-fibers lowers the CPC's ability to phosphorylate kinetochore substrates.

We have utilized a mathematical modeling approach to demonstrate that inner centromere microtubule stimulation of the CPC is fully compatible with, and in fact extends, a recently published mechano-chemical model[24]. In the previous model, the chromatin-bound pool of the CPC in the inner centromere strongly activates Aurora-B kinase because this is the place of the highest Aurora-B concentration. This pool is in a constant exchange with the soluble CPC pool, which sustains high kinase activity of the chromatin-bound CPC where it is less abundant (Fig. 9a). With increased tension, however, the concentration of chromatin-bound CPC near the kinetochore falls below the threshold for activation, so phosphorylation at the kinetochores drops despite presence of active kinase in the soluble pool, which is only 10 nM. Using this theoretical model, we tested a hypothesis that this drop in kinase activity can be prevented by centromere-proximal microtubules, enabling cells to sense inappropriate microtubule attachments (Fig. 4). Our model confirms that this scenario is feasible and the centromere-proximal microtubule bundles can propagate CPC activity, up regulating phosphorylation even at the fully stretched kinetochores without disrupting microtubule attachment at the bi-oriented kinetochores.

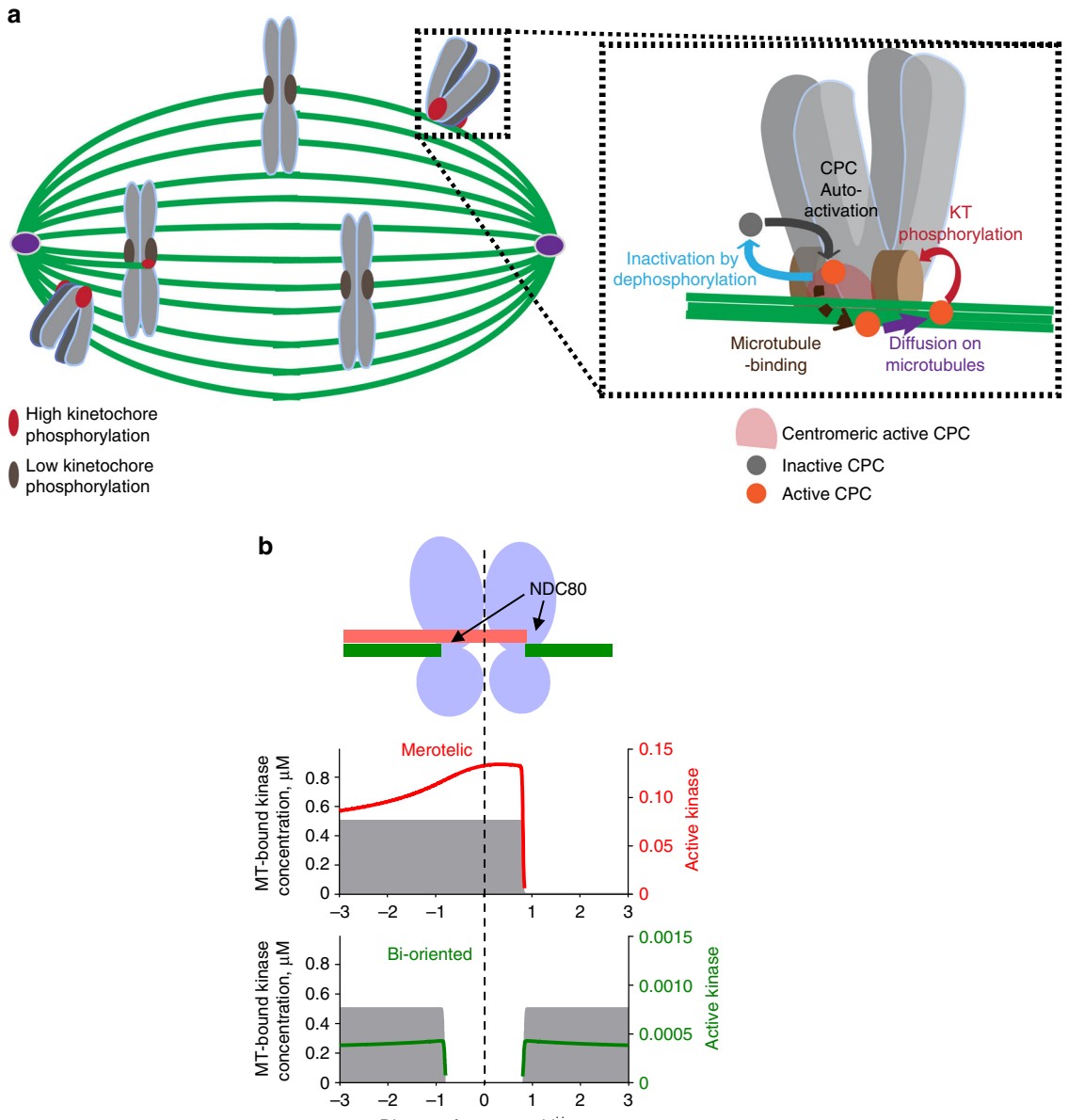

**Fig. 9** Model for kinetochore phosphorylation by the CPC. **a** Model showing phosphorylation of laterally attached kinetochore by the CPC. Inactive non-centromeric pool of CPC (gray) is auto-activated by centromeric pool (red). The non-centromeric activated CPC (orange) diffuses on centromere proximal microtubules (green) and phosphorylates kinetochores (brown). **b** Theoretical results showing predictions obtained using our model for the activity of microtubule-bound kinase at the kinetochores with merotelic microtubules (cartoon illustration on top). Graphs show concentration of the microtubule-bound kinase for merotelic and end-on microtubules (in gray, left axis). Fraction of the active microtubule-bound kinase for merotelic (red curve) and end-on microtubules (green curve) is plotted using the right axes. Kinase activity on the end-on attached microtubules is more than 100-times lower than on the merotelic ones, even though they are bound at the same kinetochore

Our mathematical model is simplified and it does not include many complexities, such as kinetochore phosphorylation by Aurora-A kinase, kinetochore-localized phosphatase or induction of CPC recruitment to centromere. Also, many parameters for the Aurora B kinase-phosphatase switch acting in cell context are not known, so the model cannot predict quantitatively the level of phosphorylation at the kinetochore. Nonetheless, our proof-of-principle model has already been highly informative because it suggests that phosphoregulation at the kinetochore relies on a highly complex biochemical system, in which chromatin-bound, microtubule-bounds and soluble CPC pools exchange dynamically. Because they are engaged in non-linear enzymatic reactions, the activities of these pools are synergistic as they amplify each

other at different locations. We show that in cells these pools are also required for high phosphorylation of prometaphase kinetochores. Thus, our data provides important in vivo support for the reaction-diffusion model of CPC activity.

Regulation of the CPC by the proximity of inner centromeres also provides insight into the long-standing question about a mechanism that enables high phosphorylation of Ndc80 complexes that are bound to the merotelic microtubules compared to amphitelic microtubules. In our model, Aurora-B bound to centromere-proximal microtubules becomes activated by centromere-bound kinase and subsequently spreads along microtubules. Because merotelic microtubules provide a path for CPC diffusion, active kinase can reach kinetochore attachment site

of these specific microtubules without affecting the ends of bi-oriented microtubules, as we illustrate by calculating profiles of active CPC at the kinetochore that has both types of microtubules (Fig. 9b).

In conclusion, we have provided mechanisms that allow the CPC to be an information processor that senses the environment around each chromosome by reading local chromatin and microtubule structures and couple this information to changes in the phosphorylation status of each kinetochore.

## Methods

**Cell culture**. HeLa T-REx cell (ThermoFisher Scientific) were grown in Dulbecco's modified Eagle's medium (DMEM, Invitrogen) supplemented with 10% fetal bovine serum (Gibco) in a humidified incubator at 37 °C in presence of 5% $CO_2$.

**Stable cell lines generation**. In order to generate HeLa T-REx cells (HeLa-TREx cells were a gift from Dan Foltz lab) stably expressing LAP-Borealin$^{WT}$, LAP-Borealin$^{MTBM}$, LAP-Borealin$^{\Delta20}$, LAP-PRC1$^{MBD}$-Borealin$^{\Delta20}$, LAP-Borealin$^{T230E}$ and LAP-Borealin$^{MTBM/T230E}$, the Borealin$^{WT}$ transgene fragment was sub-cloned in to pCDNA5/FRT vector (Invitrogen) containing N-terminal LAP (GFP and S-peptide) tag sequence. QuickChange II XL site-directed mutagenesis kit (Agilent) was used to generate all the point mutations and deletions constructs. All primers used for site-directed mutagenesis and cloning are described in Supplementary Table 1 and 2. For generating HeLa T-REx cells expressing GFP-CENP-B$^{DBD}$-INCENP$^{747–918}$ and mCherry-Borealin$^{WT}$ or mCherry-Borealin$^{MTBM}$ transgenes, the GFP-CENP-B$^{DBD}$-INCENP$^{747–918}$ was sub-cloned downstream of Tet-operator binding site. CMV promoter containing fragment of mCherry-Borealin$^{WT}$ or mCherry-Borealin$^{MTBM}$ was cloned at the 3′ end of GFP-CENP-B$^{DBD}$-INCENP$^{747–918}$. This whole cassette of GFP-CENP-B$^{DBD}$-INCENP$^{747–918}$ and mCherry-Borealin$^{WT}$ or mCherry-Borealin$^{MTBM}$ was then cloned into pCDNA5/FRT vector (Invitrogen). The resulting plasmids were co-transfected with the pOG44 plasmid (Invitrogen) with Lipofectamine 2000 (Invitrogen). Hygromycin B (Invitrogen) 200 μg/ml was added one-day post transfection and the cells were selected for 15 days. After the selection period, the surviving colonies were pooled and FACS sorted for GFP (for LAP-Borealin transgenes) or mCherry (for GFP-CENP-B$^{DBD}$-INCENP$^{747–918}$ and mCherry-Borealin$^{WT}$ or mCherry-Borealin$^{MTBM}$ transgenes) expression to get cells expressing equal amount of the transgene.

For generating cells expressing vsv-INCENP$^{WT}$-GFP or vsv-INCENP$^{\Delta SAH}$-GFP with mCherry-Borealin$^{WT}$ or mCherry-Borealin$^{MTBM}$ transgenes. The Tet-inducible HeLa T-REx cells expressing vsv-INCENP$^{WT}$-GFP or vsv-INCENP$^{\Delta SAH}$-GFP (a kind gift from S. Lens; vsv-INCENP$^{\Delta SAH}$-GFP was previously referred to as vsv-INCENP$^{\Delta SAH}$-GFP) were infected with virus carrying mCherry-Borealin$^{WT}$ or mCherry-Borealin$^{MTBM}$ transgenes in the presence of 8 μg/ml polybrene (Sigma). The double stable cells were then selected for 10–12 days in presence of Puromycin (Invitrogen) at 1 μg/ml and Hygromycin B at 200 μg/ml. The surviving colonies were pooled and FACS sorted for mCherry expression to obtain double stable cell lines.

**Virus production**. For making retrovirus, mCherry-Borealin$^{WT}$ or mCherry-Borealin$^{MTBM}$ transgenes were cloned into pBABE-Puro retrovirus vector using cold fusion cloning kit (System Biosciences). HEK-293GP cells were co-transfected with pBABE-Puro-mCherry-Borealin (WT or MTBM) and VSVG plasmid in order to package pseudotyped MULV viruses. The viruses were collected 3 days post transfection by filtering the media through 0.45 μm syringe filter.

**Plasmid and siRNA transfection and STLC washout assay**. For plasmid transfection cells were grown to 80–90% confluence followed by plasmid transfection using Lipofectamine 2000 (Invitrogen) according to the manufacture's protocol.

In case of siRNA transfection in order to avoid the indirect effect we analyzed the first mitosis after depletion of the target proteins. To achieve this, cells were plated in presence of 2 mM thymidine, 24 h after plating, cells were released into fresh media and siRNA was transfected using RNAiMAX (Invitrogen) according to the manufacturer's protocol. Another round of siRNA transfection was done after 10–12 h of 1st siRNA treatment and at the same time 2 mM Thymidine was added. Cells were released into fresh media 12h after the second siRNA treatment. For immunofluorescence analysis cells were fixed after 8–10h of second thymidine release or after the indicated treatment. Depletion of the protein upon siRNA treatment was assessed by western blotting. Uncropped scan of all the blots are shown in Supplementary Figs. 8, 9, and 10.

For STLC washout assay 6–7h after the second thymidine release 5 μM STLC was added and cells were incubated for 2h. After 2h of incubation STLC was washed out of the cells by washing with PBS followed by 1.5h of incubation at 37 °C in presence of 5% $CO_2$. Cells were fixed and stained with DAPI and mounted on coverslips using ProlongGold antifade (Invitrogen). Sequences for all the siRNA used in this study are in Supplementary Table 3. Concentrations of all the small molecule inhibitors used in this study are in Supplementary Table 4.

**Live cell imaging**. For live cell imaging, cells were plated in the 4-well Lab-Tek II chambered coverglass (Thermo Fisher Scientific) in presence 2 mM thymidine followed by siRNA treatment. After 3–4 h of the 2nd thymidine release 200 nM SiR-DNA (Cytoskeleton Inc.) dye was added to the cells. One and a half hour after SiR-DNA treatment time-lapse images were taken at 5 min interval for 15 h on a Zeiss Axio-observer-Z1 in a humidified environmental chamber maintained at 37 °C in presence of 5% $CO_2$.

**Immunoprecipitation (IP)**. For immunoprecipitation, HeLa cells were synchronized to mitosis with 0.33 μM nocodazole for 16h. Mitotic cells were collected and lysed in CPC lysis buffer (250 mM NaCl, 50 mM Tris-HCl pH 7.5, 5 mM EDTA, 0.5% NP-40, 1 mM DTT, 20 mM Beta-glycerophosphate, 50 mM NaF, 1 mM Na-orthovanadate, 1× protease inhibitors cocktail (Roche) and sonicated using Bioruptor-300 (Diagenode) for 30 cycles with 30s on and 30s off at 4 °C. The whole-cell lysate was cleared by centrifugation at 14,000 × g for 10 min and the supernatants was incubated with GFP antibody (a kind gift from Dan Foltz) for 3 h at 4 °C. Equilibrated protein-A beads (GE Life Sciences) were then incubated with the antibody lysate mixture for an additional hour. The beads were washed with lysis buffer 3 times. The washed beads were re-suspended in 2× sample buffer and loaded on to SDS–PAGE gel after brief boiling at 95 °C, desired proteins were detected in the immune-precipitate by western blotting. Uncropped images of the western blots are shown in Supplementary Fig.8.

**Immunofluoresence microscopy**. HeLa T-REx cells were seeded onto coverslips coated with poly-L-Lysine (Sigma) and indicated siRNA transfection and subsequent indicated treatment were done. The cells were then co-fixed with 4% paraformaldehyde in PHEM buffer (60 mM Pipes, 25 mM Hepes, 10 mM EGTA, and 4 mM MgCl$_2$, pH 6.9) supplemented with 0.5% Triton-X 100 for 20 min at room temperature. The cells were then washed 3 times with Tris buffered saline (TBS), followed by 1 h blocking with 3% BSA at room temperature. Fixed cells were then incubated with indicated primary antibodies for 1 h at room temperature. Concentration and source of all the antibodies used in this study are in Supplementary Table 5. After washing three times with TBS-T (TBS + 0.1% Tween20), cells were incubated with fluorescent secondary antibodies (1:2000) (Jackson Immuno-Research). After washing 4 times with TBS-T, the cells were stained with 0.5 μg/ml DAPI for 5 min and the coverslips were mounted onto slides using ProlongGold antifade (Invitrogen) and sealed with nail polish. Image acquisition was performed as described previously[35]. Images were processed and analyzed using Volocity (V5.5, PerkinElmer). To quantify fluorescence levels at centromeres, we used an intensity thresholding algorithm to mark all centromeres on the basis of ACA or GFP (for Fig. 8, Supplementary Fig. 7) intensity. To eliminate the size difference of each marked centromere the total fluorescence intensity was divided by the total volume of the selected area. Upon background subtraction the intensity/volume values of the desired channel were normalized against the corresponding ACA intensity/volume. When cells were not stained with a centromere marker or when the staining pattern was not encompassed by ACA background subtracted intensity/volume was reported. For Supplementary Fig. 2E Aurora-B intensity was quantified by circling around the Aurora-B intensity on a maximum intensity projections images using ImageJ software[61]. All the values were plotted (box and whisker plots showing 5–95% percentile (whiskers) and box representing 25–75 percentile and median is the line in the boxes) using Prism software (GraphPad) and indicated two tailed statistical tests where applied. For all the immunofluorescence experiments data from one out of atleast 2 repeats is shown.

**Protein purification**. INCENP$^{1-58}$-Survivin-Borealin complex was expressed in BL21-pLysS (DE3) cells from a tri-cistronic pET28a vector containing 6XHis-INCENP$^{1-58}$-Survivin-Borealin sequence. ISB$^{MTBM}$ and ISB$^{T230E}$ were generated by site directed mutagenesis. For GFP-ISB construct, the ISB construct was modified by sub-cloning GFP between 6His tag and INCENP$^{1-58}$. Cells were grown in 2XYT media in presence of 30 μg/ml Kanamycin. Protein expression was induced at O.D. 0.6 by addition of 0.45 mM IPTG and the media was supplemented with 0.2% glucose 60 μg/ml ZnCl$_2$, protein expression was carried out for 16–18 h at 18 °C. Cells were subsequently pelleted and lysed in buffer containing 50 mM Tris, pH 7.5; 500 mM NaCl; 0.5 mM TCEP; 5 mM Imidazole; 5% glycerol and protease inhibitor cocktail (Roche) using EmulsiFlex-C3 Homogenizer. Lysate was cleared by centrifugation at 95,834 × g for 1 h. Cleared lysate was then mixed with Ni-NTA beads (Qiagen) for 4 h at 4 °C. Ni-NTA beads (Qiagen) where then washed with 200 ml buffer containing 25 mM Imidazole, 50 mM Tris pH 7.5, 500 mM NaCl, 0.5 mM TCEP and 5% glycerol. The Protein was then eluted with buffer containing 250 mM Imidazole, 50 mM Tris pH 7.5, 500 mM NaCl, 0.5 mM TCEP and 5% glycerol. Upon elution the proteins were gel filtered on Superdex-200 column 10/300 GL size-exclusion column (GE Life Sciences). Gel filtration was done in buffer containing 50 mM Tris pH 7.5, 500 mM NaCl, 0.5 mM TCEP and 5% glycerol. Upon gel filtration the desired fractions were pooled and concentrated with Amicon Ultra-4 Centrifugal Filter Unit with 3 kDa cutoff.

**Microtubule co-sedimentation assay and microtubule bundling assay**. Taxol-stabilized microtubules were prepared by polymerizing bovine brain tubulin dimers in BRB80 (80 mM PIPES, 1 mM MgCl$_2$, 1 mM EGTA, pH 6.8 with NaOH), 50 mM

NaCl, 1 mM DTT and 1 mM GTP with increasing concentration of taxol, taxol-stabilized microtubules were then separated from the un-polymerized tubulin dimers by centrifuging through a 40% glycerol cushion at $137,000 \times g$. Various concentrations of taxol-stabilized microtubules were mixed with 100 nM of ISB (WT, MTBM or $\Delta$20) in BRB80, 1 mM DTT, 50 mM NaCl and 20 $\mu$M paclitaxel. Samples were allowed to equilibrate at room temperature for 15 min. Samples were then layered onto a 50% glycerol cushion and centrifuged at $279,000 \times g$ for 10 min, and both the supernatant (S) and pellet (P) were collected and resuspended in SDS sample buffer, and equal amounts of supernatant and pellet were run on 15% SDS-PAGE gels followed by western blotting. Quantification of the relative amounts of ISB in supernatants and pellets was performed using ImageJ (National Institutes of Health, Bethesda, MD). The dissociation constants measured by fitting the data from three separate experiments to the one-site specific binding equation using Prism software (GraphPad).

For bundling assay indicated concentrations of ISB complex were incubated with 1 $\mu$M microtubules in BRB80, 1 mM DTT, 100 mM NaCl and 20 $\mu$M paclitaxel for 15 min at room temperature. The reaction was then fixed with 1% glutaraldehyde in BRB80 for 5 min. The fixed reaction was then pipetted on the coverslips and was allowed to adhere for 10 min at room temperature. The coverslips were then blocked with 3% BSA in TBS for 30 min. The coverslips were then probed with DM1$\alpha$ and 6-His antibody in blocking solution for 1h at room temperature. After washing 3 times with TBS fluorescent secondary antibodies were added for 1h in blocking solution at room temperature. The coverslips were then washed 4 times with TBS and mounted in Prolong gold followed by imaging at 63X objective using a Zeiss Observer Z1 wide-field microscope.

**Description of mathematical model.** General model framework: Quantitative analysis of the spatial distribution of Aurora B kinase activity was carried out based on our previously published mathematical model[24]. Since the model does not discriminate between CPC and Aurora B kinase, we use these names inter-changeably throughout the theoretical section. The model incorporates (1) biochemical reactions of Aurora B phosphorylation-dependent auto-activation (in cis and in trans) and its phosphatase-dependent inactivation, (2) the reactions of Aurora B kinase binding/unbinding to the centromere-localized binding sites and (3) diffusion of the soluble kinase and phosphatase pools.

Briefly, the model is built upon the fact that Aurora-B kinase can activate itself, so its activation is strongly concentration-dependent (Fig. 4a, upper box). Biochemical analyses in vitro demonstrate that a self-activating kinase combined with an inactivating phosphatase can form a bi-stable system: almost entire kinase pool is activated when kinase concentration exceeds a threshold, whereas the kinase is mostly inactive when its concentration is below the threshold. In the model, these non-linear reactions of de/phosphorylation involve the chromatin-bound and soluble CPC pools. Because chromatin-localized CPC binding sites are enriched at the centromere (Fig. 4b), in the one-dimensional model of spatial CPC activity, the chromatin-bound CPC has a peak at the centromere and concentration decreases toward the kinetochore. Soluble CPC pool and soluble phosphatase have uniform distributions due to the fact that protein diffusion is very fast. These enzymatic components form a bi-stable spatially distributed reaction-diffusion system[24].

Spatial bi-stability is seen from the fact that kinase activity along centromere–kinetochore axis is not simply proportional to the concentration of chromatin-bound CPC. At the unstretched centromere (as in prometaphase), concentration of chromatin-bound CPC is higher than the activation threshold everywhere throughout the centromere, leading to a high level of Aurora B kinase activity at the Ndc80 site of the kinetochore (Fig. 4b). Due to bi-stability, the kinase activity drops sharply where CPC concentration is below the threshold, but at prometaphase kinetochores this boundary is located in cytoplasm (soluble CPC concentration 10 nM)[5]. To model the metaphase configuration, tension is applied to stretch the concentration profile of the chromatin-bound CPC, reducing its concentration everywhere along the centromere–kinetochore axis. This shifts the kinase activity boundary slightly closer to the centromere relative to the Ndc80 site, where CPC concentration drops below the threshold. Thus, the model recapitulates lower kinase activity at the Ndc80 site of the stretched vs. unstretched kinetochores (Fig. 4c), providing theoretical background for the tension-dependent kinetochore phosphoregulation by the CPC[24].

Additionally, we have now incorporated into this model (4) the interactions between kinase and microtubules. Two configurations were compared, in one microtubules are bound to the kinetochore end-on (bi-oriented) and in another they run adjacent to the centromere (centromere-proximal).

Main model assumptions and simplifications.

1.  We assume a rapid equilibrium in the binding-unbinding of Aurora B kinase to chromatin-localized and microtubule-localized binding sites. In our previous model, rate of kinase binding to chromatin was estimated to exceed the rate of formation of the enzyme-substrate complexes by more than three orders of magnitude. Much faster rate of kinetic binding/unbinding reactions allows considering these reactions at steady-state. After the steady-state kinase pools are defined (soluble, chromatin-bound and microtubule-bound), equations for biochemical reactions of phosphorylation/dephosphorylation are solved to determine the fraction and concentration of active kinase within each pool and the diffusion-mediated spatial profiles for these different kinase forms depending on tension or microtubule geometry.

2.  To simplify calculations, we used one-dimensional model representation along the centromere–kinetochore axis. Model parameters (Supplementary Table 6) were defined as described below to match experimental findings about Aurora B kinase levels in cells[5]. Specifically, in all calculations concentration of soluble kinase is set at 10 nM. Also, the total number of chromatin-binding and microtubule-binding sites is chosen such that the soluble pool represents ~ 17% of the total kinase (sum of all pools), similar to the 25% fraction estimated in cells[5].

3.  For enzymatic reactions we used the following rules. Soluble kinase can phosphorylate other soluble kinase molecules, as well as the chromatin-bound and microtubule-bound molecules, thereby activating them (trans-activation). The bound forms of kinase can phosphorylate soluble kinase molecules with similar enzymatic rate constants, but the catalytic rate constant for microtubule-bound kinase is assumed to be 2-fold higher (Supplementary Table 6), reflecting findings in[35]. The bound forms of the kinase can also phosphorylate each other; such activity is assumed to be 100-fold lower relative to the soluble form to account for possible steric limitations in the bound state. For simplicity, only the soluble form of phosphatase is considered; the phosphatase dephosphorylates all forms of Aurora B kinase with the same activity.

Spatial distribution profiles of kinase-binding sites: Spatial distribution of the chromatin-localized CPC-binding sites along centromere–kinetochore axis, $\text{sites}_{\text{chrom}}^{\text{total}}(x)$, is based on the experimentally measured metaphase Aurora B localization[62], similar to approach employed by[24]:

$$\text{sites}_{\text{chrom}}^{\text{total}}(\text{x}) = B_0 \cdot k / [(1 + \exp(-s(x \cdot k + \text{cent}))) \\ \cdot (1 + \exp(-s(x \cdot k - \text{cent})))]$$

where $B_0$ is the maximum concentration of CPC-binding sites at centroid (midpoint between sister kinetochores where $x = 0$). Parameter $s = 6/\mu M$ defines steepness of this profile, matching experimental distribution. Parameters cent and $k$ are used to scale the profile in response to tension: with no tension $k=1.1$ and cent $= 0.55$ (corresponding to NDC80-NDC80 distance 1.02 $\mu$m), while for the fully stretched centromere $k = 2.3$ and cent $= 0.8$ (corresponding to NDC80-NDC80 distance 1.64 $\mu$m). These NDC80 distances and therefore NDC80 locations on all graphs are based on previous findings[63].

This profile of the chromatin-localized CPC-binding sites is represented with a smoothly decaying (differentiable) function, simplifying model calculations and resulting in the "chromatin" slightly extending beyond the NDC80 site. It is unclear whether this feature of the employed profile is physiological. High resolution electron microscopy images of the kinetochore-microtubule interface in human cells show that chromatin threads often extend along the walls of the end-on attached microtubules[64]. Also, the ends of kinetochore microtubules within one kinetochore fiber do not end in the same location, spreading up to ~200 nm along this axis, so some microtubule ends appear to be surrounded by chromatin. It is unknown whether such geometry results in a significant presence of chromatin-bound CPC near the NDC80 substrate. It is also unclear whether activity of the chromatin-bound kinase toward NDC80 is similar to that of the microtubule-bound and soluble kinase forms. We note, however, that presence of the chromatin-bound kinase at the NDC80 site in our model does not affect its major conclusions. This is illustrated with Supplemental Fig. 3A, which plots the combined concentrations of active microtubule-bound kinase and active soluble kinase at the NDC80 location. Although the sum of these kinase forms is lower than when chromatin-bound kinase is included, the relative difference in total kinase levels for configurations with end-on vs. centromere-proximal microtubules is similar (compare to Fig. 4g).

The profiles of the CPC-binding sites on microtubules were different for different microtubule geometry. For the end-on attached microtubules we used:

$$MT(x) = \alpha MT^{\text{total}} / (1 + \exp(q(x_0 - x))) \\ + \alpha MT^{\text{total}} / (1 + \exp(q(-x_0 - x)))$$

where $MT^{\text{total}}$ is concentration of CPC binding sites on microtubules, $q = 100/\mu m$ is parameter that defines shape of the microtubule ends distribution, and $x_0 = 0.8$ $\mu$m is position of the microtubule ends at the stretched kinetochore, coinciding with NDC80 location. Parameter $\alpha$ describes microtubule abundance. Because our model is one-dimensional, it is difficult to ascribe specific number of microtubules to this parameter. However, this approach allows examining the relative effects of different number of kinetochore microtubules attached end-on vs. centromere-proximal. Unless stated differently, parameter $\alpha = 1$.

For centromere-proximal microtubules, we used constant level of binding sites:

$$MT(x) = \alpha MT^{\text{total}}$$

For kinetochore with both end-on and merotelic microtubules, the end-on microtubules were simulated as described above. Merotelic microtubules were described using the following profile:

$$MT(x) = MT^{\text{total}} / (1 + \exp(q(x - x_0)))$$

Calculation of the steady-state distributions of the bound kinase pools: Bound kinase pool is the sum of all chromatin- and microtubule-bound kinase forms, including active and partially active kinases, kinase complexed with phosphatase and the enzyme-substrate complexes. Steady-state one-dimensional distributions for bound kinase pools were determined using the following expressions:

$$A_{sol} + MT(x) \underset{k_{off}^{MT}}{\overset{k_{on}^{MT}}{\rightleftarrows}} MT\,Profile(x)$$

$$A_{sol} + sites_{chrom}^{free} \underset{k_{off}^{chrom}}{\overset{k_{on}^{chrom}}{\rightleftarrows}} Profile(x)$$

$$sites_{chrom}^{free} + Profile(x) = sites_{chrom}^{total}$$

where $A_{sol}$ – soluble concentration of Aurora B kinase (all forms); $MTProfile(x)$ – steady-state concentration profile of the microtubule-bound kinase (all forms); $Profile(x)$ – steady-state concentration profile of the chromatin-bound kinase (all forms); $sites_{chrom}^{free}$ is concentration profile of chromatin-binding sites that are unoccupied and available for CPC binding. Parameters $k_{on}^{MT}$, $k_{on}^{chrom}$, $k_{off}^{MT}$ and $k_{off}^{chrom}$ are association and dissociation rate constants for microtubules and chromatin.

To examine model predictions in the absence of CPC-microtubule binding, calculations were carried out as in the full model but with $k_{on}^{MT} = 0$. Likewise, modeling of CPC in the absence of chromatin binding was carried out using $k_{on}^{chrom} = 0$.

Choice of model parameters: All model parameters and their values are listed in Supplementary Table 6.

Binding constants. For Aurora B kinase interaction with chromatin, we used same parameters as used previously[24]. Parameter $B_0$, the maximum concentration of chromatin kinase binding sites at $x=0$, was set at 30 μM to match the peak concentration of chromatin-bound kinase in previous model: 20 μM. For simplicity, the association rate constant $k_{on}$ was assumed the same for microtubules and chromatin, so $k_{on}^{MT} = k_{on}^{chrom} = 2.9/\mu M/s$. The dissociation rate constant for microtubules $k_{off}^{MT} = 0.3/s$. This value was chosen based on the experimentally measured affinity of the ISB, which was ~0.1 μM (see Fig. 1a, b). With these binding constants, only 10% of the CPC-binding sites on the microtubules are occupied by CPC at steady-state. Assuming 5 μM concentration of polymerized tubulin, concentration of microtubule-bound kinase in the model is 0.5 μM, which is 20 times lower than the peak of chromatin-bound kinase (with tension). Concentration of microtubule-bound Aurora B kinase in cells is not known, but this ratio is consistent with a significantly lower fluorescence intensity of Aurora B kinase on microtubules than at the centromere[62].

Enzymatic constants: Characteristics of cytoplasmic phosphatase toward CPC are not known, so its catalytic activity was free parameter, which was adjusted to optimize model behavior. The catalytic rate of active Aurora B kinase is based on previous findings[65].

Microtubule-dependent diffusion: CPC was shown to diffuse along microtubules in yeast (Barnes) but measurements for human CPC complex were lacking. Therefore, to justify our assumption about microtubule diffusion of CPC, we examined behavior of purified ISB-GFP protein on taxol-stabilized microtubules labeled with HiLyte-645. Single molecule TIRF visualization and data analysis were carried out as previously mentioned, except motility buffer was prepared with 10mM DTT in place of beta-mercaptoethanol. Figure 1c and Supplementary Fig. 1 C-E demonstrates that this construct shows robust diffusion with diffusion coefficient 0.7 μm²/s. The diffusion rate of the full CPC, which has additional two microtubule binding sites and is likely to be dimeric, could be lower. For main model calculations we used $D_{MT} = 0.1 \ \mu m^2/s$. Additionally, we varied this coefficient to examine how the rate of CPC diffusion affects the distribution profile of the microtubule-bound active kinase. Importantly, the model remains sensitive to microtubule configuration when this coefficient ranges from 0.005 and 0.7 μm²/s (Supplementary Fig. 3C). The model also shows that in the absence of CPC diffusion, the microtubule-bound kinase is more active near the centromere but this activation fails to spread at longer distances (Supplementary Fig. 3B). Thus, CPC diffusion on microtubules is an important aspect of its mitotic function, so it would be important to investigate the diffusion of full CPC complex.

Full set of biochemical equations: The following system of differential reaction-diffusion equations was used:

$$\partial A^*/\partial t = A \cdot k_{cis} + \lfloor AA^* \rfloor \cdot (2k_{cat}^a + k_r^a) - A^* \cdot A \cdot k_f^a$$
$$- (A^* \cdot B + A^* \cdot T) \cdot k_f^a + ([AB^*] + [AT^*]) \cdot k_{cat}^a$$
$$+ ([BA^*] + [TA^*]) \cdot (k_{cat}^a + k_r^a) + [A^*PPase] \cdot k_r^p$$
$$- A^* \cdot PPase \cdot k_f^p + D \cdot \partial^2 A^*/\partial x^2$$

$$\partial B^*/\partial t = B \cdot k_{cis} + ([AB^*] + [BB^*] + [TB^*]) \cdot (k_{cat}^b + k_r^b)$$
$$- B^* \cdot A \cdot k_f^a + [BT^*] \cdot k_{cat}^T - T \cdot B^* \cdot k_f^b$$
$$- B^* \cdot B \cdot k_f^b + [BA^*] \cdot k_{cat}^a$$
$$+ [B^*PPase] \cdot k_r^p - B^* \cdot PPase \cdot k_f^p$$

$$\partial T^*/\partial t = T \cdot k_{cis} + (\lfloor TT^* \rfloor + [AT^*] + [BT^*]) \cdot (k_{cat}^T + k_r^T)$$
$$- T^* \cdot A \cdot k_f^a + [TB^*] \cdot k_{cat}^b - T^* \cdot T \cdot k_f^T$$
$$- T^* \cdot B \cdot k_f^T + [TA^*] \cdot k_{cat}^a + [T^*PPase] \cdot k_r^p$$
$$- T^* \cdot PPase \cdot k_f^p + D_{MT} \cdot \partial^2 [T^*]/\partial x^2$$

$$\partial A/\partial t = -A \cdot k_{cis} + (\lfloor AA^* \rfloor + [AB^*] + [AT^*]) \cdot k_r^a$$
$$- A \cdot (A^* + B^* + T^*) \cdot k_f^a + [A^*PPase] \cdot k_{cat}^p$$
$$+ B \cdot k_{Boff} + D \cdot \partial^2 A/\partial x^2$$

$$\partial B/\partial t = -B \cdot k_{cis} + \lfloor BA^* \rfloor \cdot k_r^a + [BB^*] \cdot k_r^b + [BT^*] \cdot k_r^T$$
$$- B \cdot A^* \cdot k_f^a - B \cdot B^* \cdot k_f^b$$
$$- B \cdot T^* \cdot k_f^T + [B^*PPase] \cdot k_{cat}^p$$

$$\partial T/\partial t = -T \cdot k_{cis} + \lfloor TA^* \rfloor \cdot k_r^a + [TT^*] \cdot k_r^T + [TB^*] \cdot k_r^b$$
$$- T \cdot A^* \cdot k_f^a - T \cdot T^* \cdot k_f^T - T \cdot B^* \cdot k_f^b$$
$$+ [T^*PPase] \cdot k_{cat}^p + D_{MT} \cdot \partial^2 [T]/\partial x^2$$

$$\partial [AA^*]/\partial t = A \cdot A^* \cdot k_f^a - \lfloor AA^* \rfloor \cdot (k_{cat}^a + k_r^a) + D \cdot \partial^2 [AA^*]/\partial x^2$$

$$\partial [BA^*]/\partial t = B \cdot A^* \cdot k_f^a - \lfloor BA^* \rfloor \cdot (k_{cat}^a + k_r^a)$$

$$\partial [AB^*]/\partial t = A \cdot B^* \cdot k_f^b - \lfloor AA^* \rfloor \cdot (k_{cat}^a + k_r^b)$$

$$\partial [BB^*]/\partial t = B \cdot B^* \cdot k_f^b - \lfloor BB^* \rfloor \cdot (k_{cat}^a + k_r^b)$$

$$\partial [TB^*]/\partial t = T \cdot B^* \cdot k_f^a - \lfloor TB^* \rfloor \cdot (k_{cat}^a + k_r^a)$$

$$\partial [TA^*]/\partial t = T \cdot A^* \cdot k_f^a - \lfloor TA^* \rfloor \cdot (k_{cat}^a + k_r^a) + D_{MT} \cdot \partial^2 [TA^*]/\partial x^2$$

$$\partial [AT^*]/\partial t = A \cdot T^* \cdot k_f^a - \lfloor AT^* \rfloor \cdot (k_{cat}^a + k_r^a) + D_{MT} \cdot \partial^2 [AT^*]/\partial x^2$$

$$\partial [TT^*]/\partial t = T \cdot T^* \cdot k_f^T - \lfloor TT^* \rfloor \cdot (k_{cat}^a + k_r^T) + D_{MT} \cdot \partial^2 [TT^*]/\partial x^2$$

$$\partial [BT^*]/\partial t = B \cdot T^* \cdot k_f^T - \lfloor BT^* \rfloor \cdot (k_{cat}^a + k_r^T)$$

$$\partial [A^*PPase]/\partial t = PPase \cdot A^* \cdot k_f^p - \lfloor A^*PPase \rfloor \cdot (k_r^p + k_{cat}^p)$$
$$+ D \cdot \partial^2 [A^*PPase]/\partial x^2$$

$$\partial [B^*PPase]/\partial t = PPase \cdot B^* \cdot k_f^p - \lfloor B^*PPase \rfloor \cdot (k_r^p + k_{cat}^p)$$

$$\partial [T^*PPase]/\partial t = PPase \cdot T^* \cdot k_f^p - \lfloor T^*PPase \rfloor \cdot (k_r^p + k_{cat}^p)$$
$$+ D_{MT} \cdot \partial^2 [T^*PPase]/\partial x^2$$

Boundary conditions were chosen to avoid the flow of soluble components:

$$\frac{dA}{dx}\bigg|_{x=0,R} = 0$$

$$\frac{dA^*}{dx}\bigg|_{x=0,R} = 0$$

$$\frac{d[AA^*]}{dx}\bigg|_{x=0,R} = 0$$

$$\frac{d[A^*PPase]}{dx}\bigg|_{x=0,R} = 0$$

$$\frac{dT}{dx}\bigg|_{x=0,R} = 0$$

$$\frac{dT^*}{dx}\bigg|_{x=0,R} = 0$$

$$\frac{d[TA^*]}{dx}\bigg|_{x=0,R} = 0$$

$$\frac{d[AT^*]}{dx}\bigg|_{x=0,R} = 0$$

$$\frac{d[T^*PPase]}{dx}\bigg|_{x=0,R} = 0$$

where $x = 0$ for the left boundary and $x = R$ for the right boundary of the simulated spatial segment. Calculations here were carried out for $R = 10\ \mu m$.

Following initial conditions were used:

$$B^*|_{t=0} = \mathrm{Profile}(x)$$

$$T^*|_{t=0} = MT\,\mathrm{Profile}(x)$$

$$\{A^*, A, B, [AA^*], [BA^*], [BB^*], [AB^*], [TA^*], [TT^*],$$
$$[AT^*], [A^*PPase], [B^*PPase], [T^*PPase]\}|_{t=0} = 0$$

Additionally, the sums of all bound and soluble kinase forms and soluble phosphatase forms were constrained:

$$\mathrm{Profile}(x) = B + B^* + 2[BB^*] + [AB^*] + [B^*PPase]$$
$$+ [TB^*] + [BT^*] + [BA^*]$$

$$MT\,\mathrm{Profile}(x) = T + T^* + 2[TT^*] + [AT^*] + [T^*PPase]$$
$$+ [TB^*] + [BT^*] + [TA^*]$$

$$A_{\mathrm{sol}} = A + A^* + 2[AA^*] + [AB^*] + [BA^*]$$
$$+ [A^*PPase] + [AT^*] + [TA^*]$$

$$PPase_0 = PPase + [A^*PPase] + [B^*PPase] + [T^*PPase]$$

Equations were solved numerically using Mathematica software (Wolfram Research) with total simulation time 50,000s using automatic time and space step size option to ensure convergence.

**Reporting summary**. Further information on experimental design is available in the Nature Research Reporting Summary linked to this article.

## Data availability

All the raw data files will be made available upon reasonable request.

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

## Acknowledgements

This work was supported by grants from the NIH to P.T.S. (1R01GM118798) and E.L.G. (R01-GM098389), NSF grant MRSEC 16 DMR-1720530 IRG:Chemo-Mechanics of Fibrous Networks, grants from the Russian Foundation for Basic Research (17-00-00480 to E.L.G. and 17-00-00481 to F.I.A.). A.V.Z is supported in part by Basic Science Program #18 from the Presidium RAS. Theoretical modeling (Fig. 4) was supported by grant from Russian Science Foundation (16-14-00-224) to F.I.A. P.T.S. and P.T. would like to thank Limin Liu and Anindya Dutta for their support, Dan Burke for critical reading of the manuscript and Dan Foltz, Arshad Desai, Susanne Lens, Jennifer Deluca, and Iain Cheeseman for reagents.

## Author contributions

P.T. performed all the biochemical and cell biological experiments under the supervision of P.T.S. except the single molecule imaging experiment, which was performed by A.V.Z. A.V.Z. and M.G. carried out theoretical analyses under the supervision of F.I.A. and E.L. G.. P.T.S., P.T., and E.L.G. wrote the paper.

## Additional information

**Competing interests:** The authors declare no competing interests.

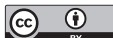

