## [Peer Review File · Nature Communications]

Reviewers' comments:

Reviewer #1 (Remarks to the Author):

Summary: The last 10 years or so has seen an intense debate between Tension-dependent and – independent models for detecting errors in kinetochore-microtubule attachment. This paper nicely positions both possibilities side-by-side and offers a somewhat unified model that combines both tension-dependent and microtubule-dependent (tension independent) mechanisms. The Borealin mutants described here offer an excellent tool to explore the tension-independent microtubule-associated role for CPC. Additionally the model presented explains how CPC can contribute to the conversion of lateral attachments into end-on ones. Hence I find the manuscript a valuable contribution to the chromosome biology field and suitable for publication in Nature Communications, provided the authors can address the following technical queries.

MAJOR COMMENTS

1. Figure 1E – What was the length of incubation time? Does this match the timepoints in the graph or MT-pull downs (1C)?

2. Figure 1E – top two right-most panels (microtubules): The 1 microMolar lane doesn't seem to fully merge with the 'merge' bottom panel (I could be wrong but I would recommend the authors to have a relook at the image placements carefully).

3. In Figure 2C, Why does LAP-Borealin appear as a doublet (two bands) but not the band marked endogenous Borealin? Is this due to protein modification? Comment in results or figure legend.

4. "Cells expressing the Borealin^{MTBM} or Borealin Δ 20 doubled the number of anaphases with lagging chromatids"

Was chromosome congression delayed similar to the anaphase onset delay? If not, there could be an unexplained silencing role as well for the microtubule-bound CPC.

5. "We conclude that the Borealin and the INCENP MBDs play different roles in the kinetochore-microtubule error correction process."

Did the double mutant localize differently compared to the single mutants? This could be useful in interpreting the phenotype.

6. Figure 5H. Is there a reduction in spindle pole associated Hec1pS69 signals?

7. "we suggest that prometaphase microtubule structures that are in immediate vicinity to the centromere, such as preformed K-fibers, or lateral attachments enable robust phosphorylation of the kinetochores⁸⁴⁻⁸⁶ (Fig. 9B)".

84 is not a relevant citation for discussing the phosphorylation of the kinetochores. In fact bridging fibres can be separated from k-fibres using diffraction-limited microscopes suggesting at least a 250nm-400nm gap between the kinetochore and the microtubule structure. While the CPC may reach out upto 500nm in the dogleash model if this microtubule structure is relevant in robust phosphorylation one would not be able to stabilize bioriented attachments in metaphase when these microtubule structures are prominent.

Minor comments

1. Minor typos in sentences:

"Cells expressing the chimeric protein resolved kinetochore-microtubule errors significantly better than the Borealin Δ 20 demonstrating that the key function of this domain is attachment to microtubules (Fig. S2A)."

"Western blots of input, supernatant (S) and pellet (P) fraction of microtubule co-sedimentation assay"

2. The introduction is a thorough and high quality review – nevertheless its running to 5 pages and could be shortened.

3. Sup Fig 4E – what does the green signal in merge refer to? Also in Sup Figure 4G and I.

Reviewer #2 (Remarks to the Author):

The identification of a microtubule binding region on Borealin is novel, of particular interest and adds a new dimension to CPC localization and function. Microtubules activate CPC, but it has not been clear how this contributes to CPC function at the inner centromeres. The current study sheds light on this process.

One prediction seems important to test: if the two histone marks are removed (5ITU + Bub1 RNAi), do the authors still see some CPC localized to centromeres especially in early prometaphase? Is the Borealin domain sufficient to localize even a small pool of CPC to the inner centromere?

An alternative interpretation is that there is no microtubule-localized pool. Instead, CPC localizes via the histone marks and once there, interacts with microtubules which enhance Aurora B activity. Enhanced Aurora B activity will enhance pH2A by activating MPS1, which further enhances CPC localization. This positive feedback loop is well documented and in principle could explain reduced localization of CPC components observed in cells expressing MTBM (Figure 7). Can the authors exclude this explanation, for example by expressing a fully constitutively active Aurora B? Or, by tethering Bub1 to kinetochores to bypass the potential feedback loop?

One of the most conspicuous feature of the CPC is relocation to midzone microtubules at anaphase, thought to be mediated by INCENP. In cells expressing MTBM, is midzone/midbody localization affected? Are cells able to complete cytokinesis, given the role of CPC in this process? The authors show that their abbreviated CPC (ISB) is able to bundle microtubules as long as the Borealin microtubule binding domain is intact. Their gel filtration experiments suggest that their CPC may exist as a dimer or oligomer. Is the bundling due to proximity of multiple microtubule binding sites provided by Borealin in a oligomeric CPC? This seems to be the most obvious interpretation since the INCENP microtubule binding regions are not present in ISB. Is microtubule bundling in this assay dependent on the dimerization domain of Borealin? I.e. does this still occur with their T230E mutant?

In their analysis of microtubule affinity, it appears that d20 has a higher affinity than the MTBM mutant – is this binding in the micromolar range meaningful or does this really reflect background in the described assay? If this is an accurate measurement, it is difficult to understand why the deletion mutant would have higher affinity for tubulin than the MTBM mutant which still contains the identified region (although in mutant form).

The model suggests that stationary pools of CPC (bound to microtubules or histones) participate in activating a soluble pool that can travel to the kinetochore. If travel is by diffusion, it seems unlikely that the relative small change in interkinetochore distance upon bipolar attachment would be able to change phosphorylation rates if the enzyme is diffusing. Also, a diffusing active kinase could target a kinetochore on a different chromosome. These difficulties may be avoided if travel is via the microtubule, however directionality would pose a problem. i.e. the plus/minus end microtubule directionality will not suffice – this would require a change in polarity at the central axis of the chromosome. As stated in the discussion, directionality could depend on H2A phosphorylation which is more abundant closer to the kinetochore. This model could be strengthened by additional analysis. For example, can CPC simultaneously bind microtubules and phosphorylated H2A? Alternatively, tethering Bub1 to CENPB should be inefficient at rescuing kinetochore phosphorylation upon knock-down of soluble Bub1. Targeting Bub1 away from the kinetochore might be expected to disrupt the localized diffusion mechanism.

The authors use a CENPB-INCENP fusion for the purpose of “increasing Aurora B at centromeres” however Aurora B levels are not analyzed under these conditions. This may be in previous papers, but it seems important here to carry out immunofluorescence to prove that indeed Aurora B kinetochore localization is increased by the fusion under conditions of their assay. Also, the authors show that kinetochores are still phosphorylated when CENPB-INCENP is combined with 5ITU and BUB1-RNAi. This observation provides good evidence for the operation of the soluble pool. However, the authors should also compare cells transfected with CENPB-INCENP and the treated with or without 5ITU/siBUB. In other words, does their CENPB-INCENP effectively replace

the chromatin-localized CPC in this experiment? The initial experiment of this series (CENPB-INCENP, 5TIU/siBUB, siBorealin) should be repeated using RNAi against Survivin to provide an independent assessment of the contribution of the soluble pool. Also, in this experiment, effects of Borealin knockdown and rescue with wild-type are shown in different panels – is the Borealin-WT-add back significantly different from the scrambled siRNA transfection (ie control for Borealin knock-down). Unless this is shown in a more comparable manner and ideally in the same experiment, it is difficult to determine how efficient the rescue was.

In figure 2C, when exactly were the protein samples collected? It is not clear from the description in the text or figure legend.

In experiments involving visual assessments (NEBD to anaphase, lagging chromosomes) were the samples quantified in a blinded manner? It would be important to use some method to minimize observer bias in these types of experiments.

In their mathematical model, the authors adjusted microtubule affinity of Borealin until CPC activity reflected known patterns. How did this theoretical affinity compare to the affinity measured using ISB? If it is greatly different, there should be some discussion of possible reasons why.

Minor points: in the text, the authors refer to a SAH mutant of INCENP, in the figure it appears to be designated “CC”. It would be easier to follow with the same naming.

Last sentence of page 13 is having some grammar issues.

Reviewer #3 (Remarks to the Author):

The authors investigate the role of the chromosome passenger complex in regulating kinetochore Mt attachments using experiments and modeling. The question is how regulation of phosphorylation at the kinetochore can be regulated by Aurora B kinase in the CPC that is located ~500 nm away on centromeric DNA. In a previous eLife paper, the Grischuk group developed a reaction-diffusion model that characterized Aurora B activation and suggested that a reaction-diffusion type of mechanism could account for the activity. Here, based on a new microtubule binding domain on Borealin, the authors propose that there is a centromeric DNA-bound pool of CPC, a microtubule-bound pool, and a soluble pool of CPC. The result is that when a centromeric-proximal microtubule is present (as happens early on or with improper attachments), the kinase activity at the Ndc80 location is enhanced, which would then decrease the Ndc80-microtubule affinity.

There is no escaping that the model is complicated and the parameters not tightly constrained by experiments. However, the authors are reasonably conservative about posing the model as a proof of principle rather than a quantitative accounting of the mechanism. And the mechanism is complicated, making simple models insufficient. Also, the model is reasonably well incorporated with the experimental data, and they synergize. Finally, an earlier version of the model is published in a nice eLife paper, which strengthens the model as a useful tool to be applied to the problem at hand.

In trying to intuit what is happening in the model, I concluded that, because the Aurora B kinase activation is so nonlinear due to the trans-activation, then including a microtubule that binds the kinase to the existing pool of centromere-bound kinase tips the scale so that the active kinase concentration spreads out considerably and overlaps with the Ndc80 site. Thus, the system will be very sensitive to small parameter changes. It would be possible to do a parameter sensitivity analysis, but I don't think that would add much because of the “proof-of-concept” nature of the model.

The model took a long time for me to understand. I wish the fig 9 diagrams showing the soluble,

activated kinase were in fig 4 because it is not immediately clear to the reader that the key is the centromere- and microtubule-bound kinase is activating the soluble kinase, and the soluble is phosphorylating the Ndc80. This point should be made graphically better than it is. I didn't like the 4B-D plots, it took me a long time to comprehend them; the authors should show on the two y-axes which line they refer without the reader having to get it from the legend. Also, on x-axis in fig 4G and H, the authors should specify centromere-proximal microtubules, not all (kinetochore bound) microtubules (assuming I have this right).

Overall, I think the authors need to better describe the model for readers to have a chance to understand it. If I'm not mistaken, the key point is that centromere-proximal microtubules increase the local Aurora B concentration and because of the positive feedback of activation, this has an outsized effect in spreading the range of active, soluble kinase. Making that point clear would help.

Reviewer #1 (Remarks to the Author):

Summary: The last 10 years or so has seen an intense debate between Tension-dependent and – independent models for detecting errors in kinetochore-microtubule attachment. This paper nicely positions both possibilities side-by-side and offers a somewhat unified model that combines both tension-dependent and microtubule-dependent (tension independent) mechanisms. The Borealin mutants described here offer an excellent tool to explore the tension-independent microtubule-associated role for CPC. Additionally the model presented explains how CPC can contribute to the conversion of lateral attachments into end-on ones. Hence I find the manuscript a valuable contribution to the chromosome biology field and suitable for publication in Nature Communications, provided the authors can address the following technical queries.

We thank the reviewer for appreciating the work and highlighting its importance.

MAJOR COMMENTS

1. Figure 1E – What was the length of incubation time? Does this match the timepoints in the graph or MT-pull downs (1C)?

Incubation time was 15min as mentioned in the materials and methods. We have now included this information in the figure legends also.

2. Figure 1E – top two right-most panels (microtubules): The 1 microMolar lane doesn't seem to fully merge with the 'merge' bottom panel (I could be wrong but I would recommend the authors to have a relook at the image placements carefully).

We thank the reviewer for pointing out this error. It has been corrected.

3. In Figure 2C, Why does LAP-Borealin appear as a doublet (two bands) but not the band marked endogenous Borealin? Is this due to protein modification? Comment in results or figure legend.

This is a common non-specific band that is seen with a number of Borealin antibodies made against the whole protein, which is also the case with some batches of our antibody. We have shown a western blot of cells expressing LAP-Borealin and the parent unmodified HeLa-Trex to illustrate this point in supplement figure 2F.

4. "Cells expressing the Borealin^{MTBM} or Borealin^{Δ20} doubled the number of anaphases with lagging chromatids"

Was chromosome congression delayed similar to the anaphase onset delay? If not, there could be an unexplained silencing role as well for the microtubule-bound CPC.

Good point, we have included data showing a modest increase in NEBD to Metaphase duration in the mutant cells compared to the wild type cells (Supplement figure. 2A). We have added the additional conclusion that cells expressing the mutant have a defect in chromosome congression.

We directly measured the time from metaphase alignment to anaphase and could only measure a very small increase in the LAP-Borealin^{Δ20}. This was the only phenotype not shared with the LAP-Borealin^{MTBM} which showed no change (Fig S2A). Therefore, we don't think the microtubule binding domain has a role in SAC silencing.

5. "We conclude that the Borealin and the INCENP MBDs play different roles in the kinetochore-microtubule error correction process."

Did the double mutant localize differently compared to the single mutants? This could be useful in interpreting the phenotype.

Good point. We have now included the Aurora-B localization data for these mutants in Supplement Figure 2E. We observed that individually the INCENP^{ΔSAH} and Borealin^{MTBM} expressing cells had a ~50% reduction in the amount of the CPC in the inner-centromere compared to the INCENP and Borealin WT expressing cells. This was consistent with our data in this manuscript on Borealin^{MTBM} mutant and previously published work on INCENP^{ΔSAH} mutant. We observed around ~10 percent further reduction in the levels of the centromeric CPC in cells co-expressing INCENP^{ΔSAH} and Borealin^{MTBM} compared to cells expressing either INCENP^{ΔSAH} or Borealin^{MTBM}. Although its possible that this extra reduction of the CPC in the inner-centromere, in cells co-expressing INCENP^{ΔSAH} and Borealin^{MTBM} compared to cells expressing either INCENP^{ΔSAH} or Borealin^{MTBM}, may underlie the dramatic increase in the cells with anaphase lagging chromosome. We think a more likely explanation is that the almost complete loss of microtubule binding in the double mutant cells underlies this dramatic phenotype.

6. Figure 5H. Is there a reduction in spindle pole associated Hec1pS69 signals?

We observed no significant difference between Borealin WT and MTBM expressing cells after quantifying the pole staining of the Hec1pS69 antibody. We have exchanged the image in figure 5H to show a more representative image.

7. "we suggest that prometaphase microtubule structures that are in immediate vicinity to the centromere, such as preformed K-fibers, or lateral attachments enable robust phosphorylation of the kinetochores⁸⁴⁻⁸⁶ (Fig.9B)".

84 is not a relevant citation for discussing the phosphorylation of the kinetochores. In fact bridging fibres can be separated from k-fibres using diffraction-limited microscopes suggesting at least a 250nm-400nm gap between the kinetochore and the microtubule structure. While the CPC may reach out upto 500nm in the dogleash model if this microtubule structure is relevant in robust phosphorylation one would not be able to stabilize bioriented attachments in metaphase when these microtubule structures are prominent.

Good point and we have extensively modified the text to incorporate this point in our manuscript.

Minor comments

1. Minor typos in sentences:

"Cells expressing the chimeric protein resolved kinetochore-microtubule errors significantly better than the Borealin^{Δ20} demonstrating that the key function of this domain is attachment to microtubules (Fig. S2A)."

"Western blots of input, supernatant (S) and pellet (P) fraction of microtubule co-sedimentation assay"

Thanks for pointing out the error. We have corrected these typos in the revised manuscript.

2. The introduction is a thorough and high quality review – nevertheless its running to 5 pages and could be shortened.

We have shortened the introduction, without removing any important information.

3. Sup Fig 4E – what does the green signal in merge refer to? Also in Sup Figure 4G and I.

Green signal refers to LAP-Borealin signal. We have included this information in the figure legends.

Reviewer #2 (Remarks to the Author):

The identification of a microtubule binding region on Borealin is novel, of particular interest and adds a new dimension to CPC localization and function. Microtubules activate CPC, but it has not been clear how this contributes to CPC function at the inner centromeres. The current study sheds light on this process.

We thank the reviewer for highlighting the importance of the work.

One prediction seems important to test: if the two histone marks are removed (5ITU + Bub1 RNAi), do the authors still see some CPC localized to centromeres especially in early prometaphase?

Good point, 5ITU + Bub1 RNAi treatment reduced the Aurora-B at the inner-centromere to essentially background levels as shown in supplementary figures 7A, B. This experiment was done in the absence of microtubules, which allows us to look at CPC that is localized to chromatin and avoids the misinterpretation of results that might arise due to lack of spatial resolution of light microscopy in differentiating CPC at the inner-centromere vs on the microtubules near inner-centromere.

Whenever we have treated cells with the 5ITU + Bub1 RNAi (Fig. 8 and Sup. Fig. 7) we have included the Aurora-B levels in the figure to demonstrate that the Aurora-B levels at the centromeres are indeed similar in the tested conditions.

Is the Borealin domain sufficient to localize even a small pool of CPC to the inner centromere?

We appreciate this concern and thus in order to investigate if the Borealin microtubule binding region is sufficient for localizing some CPC to the inner-centromere, we assessed the amount of Aurora-B left at the inner-centromere in cells rescued with either LAP-Borealin^{WT} or LAP-Borealin^{MTBM} upon treatment with siBub1 and 5ITU, the results are shown in Supplement figure 7C, D. Consistent with our previous observation upon siBub1 and 5ITU treatment the amount of the CPC at the inner-centromere dramatically reduced in both the cells rescued with either LAP-Borealin^{WT} or LAP-Borealin^{MTBM}. The amount of CPC left at the inner-centromere upon siBub1 and 5ITU treatment was not reduced in cells rescued with LAP-Borealin^{MTBM} compared to cells rescued with LAP-Borealin^{WT}. We thus conclude that the microtubule-binding region of the CPC is not sufficient to localize any CPC to the inner-centromere.

An alternative interpretation is that there is no microtubule-localized pool. Instead, CPC localizes via the histone marks and once there, interacts with microtubules which enhance Aurora B activity.

To clarify, when CPC “interacts with microtubules”, we refer to this CPC as a microtubule-bound (or microtubule-localized) pool. Because CPC interactions with both microtubules and chromatin are dynamic, these pools are constantly exchanging and there is a binding-unbinding equilibrium. In this sense, the interpretation proposed by this reviewer is not alternative, because it relies on CPC interaction with microtubules (i.e. binding, diffusion and unbinding) and on CPC activation when it is bound to the chromatin and microtubules.

Enhanced Aurora B activity will enhance pH2A by activating MPS1, which further enhances CPC localization. This positive feedback loop is well documented and in principle could explain reduced localization of CPC components observed in cells expressing MTBM (Figure 7). Can the authors exclude this explanation, for example by expressing a fully constitutively active Aurora B? Or, by tethering Bub1 to kinetochores to bypass the potential feedback loop?

We fully appreciate this concern, which drove the experiments in figure 6A,B and Sup. Fig. 6A,B. First we asked whether the role of the microtubule-binding domain is in activating the CPC, but we could not measure any role. The reduction in Aurora-B protein amount at the inner-centromere is similar to the reduction of the active Aurora-B (assessed by staining with Aurora-B pT232 antibody) when we compared cells expressing the wild type MTBM Borealin (Fig. 6A,B and Sup. Fig. 6A,B). This observation is inconsistent with the role of microtubules in activating H2A-pT120 bound CPC, which would predict that the Aurora-B pT232 levels at the inner-centromere would be reduced more than the levels of Aurora-B in the inner-centromere.

Second we found that microtubule binding was required at the step between inner centromere localization and kinetochore phosphorylation. To do this we established conditions where the endogenous CPC cannot localize to chromatin but there is a pool of active

centromeric Aurora B that does not recruit borealin (cells expressing CENP-B^{DBD}-INCENP⁷⁴⁷⁻⁹¹⁸ in cells treated with Bub1siRNA and 5ITU). Under this condition the microtubule binding activity of borealin is still critical for kinetochore phosphorylation as shown in Fig. 8F-I and Sup. Fig. 7 N,O. This experiment demonstrates a clear requirement for both centromere targeted and noncentromeric targeted pools in phosphorylating kinetochores. In response to the reviewers concerns we have added more data to demonstrate that the endogenous CPC is not localizing to chromatin in our experiment. Specifically we confirmed that after 5ITU and Bub1 siRNA treatment there is little to no H2ApT120 remaining and the endogenous CPC cannot localize to the chromatin (Supplement figure 7A-E). Together our observations suggest that a nonchromatin bound pool of the CPC must be able to interact with microtubules to enable kinetochore phosphorylation. While there may be a small role for the microtubules in the activation of the centromere pool our data suggest that this is not the major reason for reduction in kinetochore phosphorylation.

The Bub1 targeting experiment suggested is an interesting experiment. In our opinion it will be tough to interpret because of following reasons:

1. It is an assumption that the only role Mps1 plays in the feedback loop is to localize Bub1 to the kinetochore. Mps1 might have a Bub1 independent role in localizing CPC to the centromere, as was suggested in a "Cell" paper from the Kops lab (Jelluma. et.al., Cell, 2008). If this is the case Bub1 targeting will not properly rescue the CPC localization.
2. Even if the Bub1 targeting is sufficient to increase the amount of CPC to the inner-centromere, to find conditions where Bub1 localization would make the amount of the CPC similar between WT and MTBM mutants will not be trivial.

Given these caveats with Bub1 targeting experiments we think the experiment of removing endogenous CPC targeting signals and restoring with CENP-B^{DBD}-INCENP⁷⁴⁷⁻⁹¹⁸ is easier to interpret for two reasons. First it directly restores CPC to centromeres, which eliminates any caveats generated by unappreciated steps in the feedback from kinetochores to the centromere. Second, the removal of the SAH domain and the rest of the N-terminal INCENP, allowed us to rule out stretching of INCENP from the centromere to phosphorylate kinetochores (i.e. the "dog-leash" model). Thus we were able to specifically look at role of the non-centromeric CPC in kinetochore phosphorylation.

One of the most conspicuous feature of the CPC is relocalization to midzone microtubules at anaphase, thought to be mediated by INCENP. In cells expressing MTBM, is midzone/midbody localization affected? Are cells able to complete cytokinesis, given the role of CPC in this process?

The anaphase localization of the CPC is indeed a very interesting topic. We note that the anaphase localization and cytokinesis phenotypes of the MTBM have been published before, although their role as a microtubule binding activity was not appreciated. The Conti group identified this region of Borealin as highly conserved and showed that the mutant was unable to localize to midzones and the cells failed cytokinesis (Jeyaprakash et. al., Cell, 2007).

When we found the mutations affected an unappreciated microtubule binding activity on Borealin, we decided to concentrate on the prometaphase roles of the microtubule binding since the anaphase requirements were already published. It is rather obvious why a loss of a microtubule binding activity would affect anaphase and cytokinesis, since it is well established that the CPC binds midzones. In our opinion further analysis of anaphase is therefore beyond the scope of this paper. Instead we have focused on the prometaphase roles, which in our opinion was more mysterious and novel, since it is more difficult to see the microtubule bound pools. However, we note that we have previously published that the CPC can specifically bind preformed K-fibers in prometaphase spindles (Banerjee et al., JCB, 2014), which is in perfect agreement with this current study.

The authors show that their abbreviated CPC (ISB) is able to bundle microtubules as long as the Borealin microtubule binding domain is intact. Their gel filtration experiments suggest that their CPC may exist as a dimer or oligomer. Is the bundling due to proximity of multiple microtubule binding sites provided by Borealin in a oligomeric CPC? This seems to be the most obvious interpretation since the INCENP microtubule binding regions are not present in ISB. Is microtubule bundling in this assay dependent on the dimerization domain of Borealin? Does this still occur with their T230E mutant?

We agree that the bundling is probably mediated by oligomerization. However, this does not seem to be mediated by the dimerization domain. When we tested the ISB T230E, the bundling activity persisted (Fig. S1F). Thus, it is currently unclear how the bundling activity is generated and therefore we have not emphasized this activity, but simply shown the result.

In their analysis of microtubule affinity, it appears that d20 has a higher affinity than the MTBM mutant – is this binding in the micromolar range meaningful or does this really reflect background in the described assay? If this is an accurate measurement, it is difficult to understand why the deletion mutant would have higher affinity for tubulin than the MTBM mutant which still contains the identified region (although in mutant form).

Careful examination of the MTB region in the crystal structure shows that a positive charged region contains amino acids from INCENP as well as the amino acids on borealin that we have mutated here. Thus it is possible that the measurement is accurate and the charge reversal used in the MBTM is dominantly inhibiting some microtubule binding affinity that remains from adjacent positively charged residues on the INCENP subunit. However we note that MT pelleting assay is a good assay for inferring gross MT binding affinity, but it is unreliable for interpreting subtle changes in MT affinity since it is a non-equilibrium assay. Therefore the affinity difference may also reflect experimental uncertainty, especially since the maximum ISB bound to the microtubules between the two mutants is not that different.

The model suggests that stationary pools of CPC (bound to microtubules or histones) participate in activating a soluble pool that can travel to the kinetochore. If travel is by diffusion, it seems unlikely that the relative small change in interkinetochore distance upon bipolar attachment would be able to change phosphorylation rates if the enzyme is diffusing. Also, a diffusing active kinase could target a kinetochore on a different chromosome.

The system is very complex and not intuitive. In fact this was demonstrated in the Zytsev paper where we showed that, because of the nonlinearity of the system, small changes to kinetochore distance can generate dramatic changes to kinetochore phosphorylation. The nonlinearity arises because the kinase autoactivates and becomes inactivated by a phosphatase in accordance to Michaelis-Menten kinetics, so these reactions lead to a different level of kinase activity depending on spatial localization and local concentration of all components. This is why we use mathematical model to calculate changes in enzyme phosphorylation rates (kinase activity) at kinetochore, and how it changes when interkinetochore distance increases two-fold from prometaphase to metaphase. To make this model more accessible, we rewrote the theoretical methods section to better explain how the published model was built from first principles (Zytsev et al., ELife 2016). Most importantly, the CPC activation in the model relies strongly on the chromatin-bound kinase, which forms a gradient of active kinase extending all the way to the kinetochore. While soluble kinase is involved in shaping this gradient and in phosphorylating kinetochore targets, our model shows that it does not “travel” long distances (from centromere to kinetochore) as originally envisioned by the diffusion-based “substrate-separation” model. Instead, the CPC generates a bistable chromatin-associated media, which enables changes in active aurora-B concentration at kinetochores in response to the inter-kinetochore distance (Fig. 8 in Zytsev et al., 2016).

In the current work we have extended this model to include the effects of microtubule binding by the CPC. The system is even more complex because the kinase can diffuse through cytoplasm, and it also diffuses along microtubules. Surprisingly, the model predicts that only microtubules localized close to the inner centromere would induce robust kinetochore phosphorylation, whereas microtubules that localize outside this region have little effect. To further clarify this point, we have added new calculation. We considered the case when merotelic and bi-oriented microtubules are attached to the same kinetochore (Figure 9B). Because in our model, the kinase activity propagates along microtubules (via diffusion of already active kinase and also via a “relay” mechanism, when active kinase diffuses and activates another kinase, etc.), the CPC activity at the kinetochore is high only on the merotelic but not bi-oriented microtubules (new Figure 9B). This explains both, how CPC discriminates inappropriate microtubules, and by extension, why the microtubule binding by the CPC does not disrupt the chromosome autonomous signaling. Unlike all previous proposals to explain how CPC discriminates inappropriate microtubules, our model is backed up by quantitative

theoretical foundation, so we think this addition strongly enhances the impact of our work, and thank the reviewer for raising this important question.

These difficulties may be avoided if travel is via the microtubule, however directionality would pose a problem. i.e. the plus/minus end microtubule directionality will not suffice – this would require a change in polarity at the central axis of the chromosome.

In our model we assumed that CPC can diffuse along microtubules, which is fundamentally non-directional type of motion, so it does not depend on microtubule polarity. To further justify this assumption we have carried out single molecule analysis of ISB-microtubule interactions and found that indeed GFP-ISB exhibits random linear diffusion (see new Figure 1C S1C-E).

As stated in the discussion, directionality could depend on H2A phosphorylation which is more abundant closer to the kinetochore. This model could be strengthened by additional analysis. For example, can CPC simultaneously bind microtubules and phosphorylated H2A? Alternatively, tethering Bub1 to CENPB should be inefficient at rescuing kinetochore phosphorylation upon knock-down of soluble Bub1. Targeting Bub1 away from the kinetochore might be expected to disrupt the localized diffusion mechanism.

This is an interesting idea, and it is likely that the CPC could simultaneously bind microtubules and phosphorylated histones (since they are on distinct regions of the complex). However, we feel that it is difficult to test this model (see the argument for the complications of targeting Bub1 above) and we have provided two lines of independent evidence for a much simpler model.

First, we see robust borealin-microtubule binding dependent phosphorylation of kinetochores after eliminating Bub1 and Haspin activity (figure 8). In this set of experiments there is no Histone H2A phosphorylation or centromere bound borealin (and hence gradient of chromatin-bound CPC), yet kinetochore phosphorylation is responsive to the microtubule-binding domain on borealin. In response to your concern we have included data in the revision to show that histone H2ApT120 is eliminated in these experiments (Supplement figure 7E).

Second, the model predicts a robust response in kinetochore phosphorylation without having to invoke directionality of CPC motion, such that could be provided by microtubule-dependent motors or microtubule de/polymerization. It does not rule out that CPC can somehow travel directionally along microtubules, but we feel that this assumption is not needed and would be criticized for not being sufficiently justified.

The authors use a CENPB-INCENP fusion for the purpose of “increasing Aurora B at centromeres” however Aurora B levels are not analyzed under these conditions. This may be in previous papers, but it seems important here to carry out immunofluorescence to prove that indeed Aurora B kinetochore localization is increased by the fusion under conditions of their assay.

We agree with the reviewer and have assessed the amount of Aurora-B at the centromeres under these conditions and observe that the CENP-B^{DBD}-INCENP⁷⁴⁷⁻⁹¹⁸ targeting indeed increases the amount of Aurora-B in the inner-centromeres (Supplement figure 2G).

Also, the authors show that kinetochores are still phosphorylated when CENPB-INCENP is combined with 5ITU and BUB1-RNAi. This observation provides good evidence for the operation of the soluble pool. However, the authors should also compare cells transfected with CENPB-INCENP and the treated with or without 5ITU/siBUB. In other words, does their CENPB-INCENP effectively replace the chromatin-localized CPC in this experiment?

Good point. In order to answer the reviewers concern we either left cells untreated or treated them with siBub1/5ITU and quantified the amount of Hec1pS55 in these conditions. We observed ~50-55% reduction in the amount of Hec1 phosphorylation upon treatment with siBub1/5ITU. Upon expression of CENP-B^{DBD}-INCENP⁷⁴⁷⁻⁹¹⁸ in cells treated with siBub1/5ITU we observed an almost complete rescue of the Hec1 phosphorylation. We thus conclude that the CENP-B^{DBD}-INCENP⁷⁴⁷⁻⁹¹⁸ can effectively replace endogenous CPC under the conditions tested. The data is shown in Supplement figure 7F,G. Note that the ~50% reduction upon siBub1/5ITU

treatment is consistent with fact that Aurora-B is only partially responsible for phosphorylation of Hec1 S55 the rest is contributed by Aurora-A (Deluca et.al.,JCB,2017).

The initial experiment of this series (CENPB-INCENP, 5TIU/siBUB, siBorealin) should be repeated using RNAi against Survivin to provide an independent assessment of the contribution of the soluble pool.

In order to provide an independent assessment of the contribution of the soluble pool of CPC in phosphorylating Hec1, we treated the CENP-B^{DBD}-INCENP⁷⁴⁷⁻⁹¹⁸ expressing cells with 5TIU/siBub1 and then either treated them with Luciferase or with INCENP siRNA (targeting the 3'UTR) and assessed the amount of phosphorylation at Hec1 S55 site. Similar to our observations with borealin knockdown we observed that INCENP knockdown also reduced the amount of Hec1pS55 by ~50% compared to control knockdown (Supplement figure 7H,K, and L. Thus, we demonstrate the contribution of non-centromeric pool of CPC in phosphorylating the kinetochore substrates using two independent methods to knock down the non-centromere targeted pool.

Also, in this experiment, effects of Borealin knockdown and rescue with wild-type are shown in different panels – is the Borealin-WT-add back significantly different from the scrambled siRNA transfection (ie control for Borealin knock-down). Unless this is shown in a more comparable manner and ideally in the same experiment, it is difficult to determine how efficient the rescue was.

We appreciated reviewers concern and thus performed the suggested experiment, which is shown in supplement figure 7N,O. We observed that upon knocking down Borealin in CENP-B^{DBD}-INCENP⁷⁴⁷⁻⁹¹⁸ targeted and 5TIU/siBub1 treated cells we observed a ~50% reduction of Hec1pS55 staining, which was rescued to ~75% with the expression of mCherry-Borealin^{WT}. The expression of mCherry-Borealin^{MTBM} under these conditions was not as effective in rescuing and the Hec1pS55 staining was only rescued to ~55%. We thus maintain our original conclusion that the non-centromeric CPC needs to interact with microtubules in order to efficiently phosphorylate Hec1.

In figure 2C, when exactly were the protein samples collected? It is not clear from the description in the text or figure legend.

We have clarified this in the revised manuscript.

In experiments involving visual assessments (NEBD to anaphase, lagging chromosomes) were the samples quantified in a blinded manner? It would be important to use some method to minimize observer bias in these types of experiments.

Upon blind scoring of at least one repeat of all the movies we saw no significant difference compared to the reported values, we have thus retained the original quantifications.

In their mathematical model, the authors adjusted microtubule affinity of Borealin until CPC activity reflected known patterns. How did this theoretical affinity compare to the affinity measured using ISB? If it is greatly different, there should be some discussion of possible reasons why.

To address this point we have improved our model to incorporate this measured affinity into our model (see table 1 that lists model parameters), and we show same conceptual model behavior. We note in text that affinity of the full CPC might be different than that of ISB because CPC has additional microtubule-binding sites. Because this and many other parameters for CPC activity in cells are not known, our model cannot provide exact quantitative description of these processes. Instead, we treat it as a proof-of principle model that demonstrates feasibility and physical soundness of our hypothesis.

Minor points: in the text, the authors refer to a SAH mutant of INCENP, in the figure it appears to be designated "CC". It would be easier to follow with the same naming.

Good point. We have corrected this issue.

Last sentence of page 13 is having some grammar issues.

This has been corrected.

Reviewer #3 (Remarks to the Author):

The authors investigate the role of the chromosome passenger complex in regulating kinetochore Mt attachments using experiments and modeling. The question is how regulation of phosphorylation at the kinetochore can be regulated by Aurora B kinase in the CPC that is located ~500 nm away on centromeric DNA. In a previous eLife paper, the Grischuk group developed a reaction-diffusion model that characterized Aurora B activation and suggested that a reaction-diffusion type of mechanism could account for the activity. Here, based on a new microtubule binding domain on Borealin, the authors propose that there is a centromeric DNA-bound pool of CPC, a microtubule-bound pool, and a soluble pool of CPC. The result is that when a centromeric-proximal microtubule is present (as happens early on or with improper attachments), the kinase activity at the Ndc80 location is enhanced, which would then decrease the Ndc80-microtubule affinity.

There is no escaping that the model is complicated and the parameters not tightly constrained by experiments. However, the authors are reasonably conservative about posing the model as a proof of principle rather than a quantitative accounting of the mechanism. And the mechanism is complicated, making simple models insufficient. Also, the model is reasonably well incorporated with the experimental data, and they synergize. Finally, an earlier version of the model is published in a nice eLife paper, which strengthens the model as a useful tool to be applied to the problem at hand.

In trying to intuit what is happening in the model, I concluded that, because the Aurora B kinase activation is so nonlinear due to the trans-activation, then including a microtubule that binds the kinase to the existing pool of centromere-bound kinase tips the scale so that the active kinase concentration spreads out considerably and overlaps with the Ndc80 site.

This is exactly right and to strengthen this point we now provide a new graph with these activity profiles (Fig. 4 D-E).

Thus, the system will be very sensitive to small parameter changes. It would be possible to do a parameter sensitivity analysis, but I don't think that would add much because of the "proof-of-concept" nature of the model.

Our preliminary analysis shows that the conceptual model behavior is fairly robust. For example, we have now constrained some model parameters, e.g. by including experimentally measured ISB affinity to microtubules. Although the predicted absolute kinase activity values have changed, the model still shows nicely different response to end-on vs. centromere proximal microtubules (Fig 4G). As reviewer pointed out, there are many model parameters that cannot be constrained due to a lack of experimental measurements. We agree with the reviewer's assessment that a rigorous parameter sensitivity analysis is premature and will not be productive. The current value of this model is that it demonstrates that the mechanism that we propose is physically plausible for parameter values within the biologically reasonable range. More detailed investigation will have to await additional experimental quantifications.

The model took a long time for me to understand. I wish the fig 9 diagrams showing the soluble, activated kinase were in fig 4 because it is not immediately clear to the reader that the key is the centromere- and microtubule-bound kinase is activating the soluble kinase, and the soluble is phosphorylating the Ndc80. This point should be made graphically better than it is.

The reviewer is right about the important role played by soluble kinase (as shown in Fig9 diagrams), but the reaction network is much more complicated. For example, in cells, both chromatin-bound and microtubule-bound kinases are present at the Ndc80 location and can potentially phosphorylate Ndc80. We thoroughly rewrote theoretical section to make it more clear that establishment of the spatial gradient of CPC activity requires all three pools, and that microtubule-bound and chromatin bound kinases can also activate each other. We provide further evidence that all pools contribute to the establishment of activity gradients in a very

complex, synergistic and sometimes non-intuitive manner (Fig. 4 G and H). This synergistic spatially-distributed biochemical reactions (binding/unbinding, activation/inactivation, soluble and microtubule-dependent diffusion) collectively result in the presence of active kinase from all three pools at the Ndc80 site. Since the activity of these kinase forms toward Ndc80 is not known, the simplest assumption we use is that they each can phosphorylate Ndc80, not just the soluble kinase. These TOTAL kinase levels are plotted in Fig 4G, which shows that the model predicts significant kinase activity at the Ndc80 site in the absence of soluble kinase (see Fig. 4G last column. To make this point more clear, we additionally provide a graph that shows a sum of only two kinase forms (soluble and microtubule-bound), see Sup Fig 3A; although the total kinase level is decreased when chromatin-bound kinase is assumed to be incapable for phosphorylating Ndc80 (eg due to steric limitations), the remaining kinase pools are still different at merotelic vs end-on attached microtubules, so the model is still sensitive to microtubule configuration. Indeed, this is a very complex model, which we built using current knowledge of the spatial distribution and concentration of different kinase forms. We hope that our improved text and new figures will clarify model's underpinnings.

I didn't like the 4B-D plots, it took me a long time to comprehend them; the authors should show on the two y-axes which line they refer without the reader having to get it from the legend. Also, on x-axis in fig 4G and H, the authors should specify centromere-proximal microtubules, not all (kinetochore bound) microtubules (assuming I have this right).

We replaced these plots with a different set of graphs which use additional labels and color-coding. Legend to x-axis (current panel H) has been changed. Indeed, this panel shows results for configuration in which only the centromere-proximal microtubules are present.

Overall, I think the authors need to better describe the model for readers to have a chance to understand it. If I'm not mistaken, the key point is that centromere-proximal microtubules increase the local Aurora B concentration and because of the positive feedback of activation, this has an outsized effect in spreading the range of active, soluble kinase. Making that point clear would help.

We rewrote and expanded model description (in Methods), main text and discussion. The key point is the unexpected importance of the microtubule-bound kinase pool, which allows discriminating proper and improper microtubules by the CPC. Specifically, we show that it is physically possible for this system to have low kinase activity at the Ndc80 bound to the end-on microtubules and the relatively high kinase activity at the Ndc80 bound to merotelic (or centromere-proximal) microtubules despite the fact that CPC is assumed to bind with the same affinity all types of microtubules.

REVIEWERS' COMMENTS:

Reviewer #1 (Remarks to the Author):

The paper aims to separate tension dependent and independent roles of CPC phosphoregulation. By combining experimental and modelling data, the manuscript presents an original findings that kinetochore phosphorylation is greatly enhanced when CPC binds microtubules interacting with the centromere - a common feature of lateral microtubules in prometaphase.

The rewrite has improved the readability of the manuscript, including the modelling part of the manuscript. The proof-of-principle model is a good addition to the experimental work, as it provides a new explanation for how CPC may discriminate immature microtubule attachments that dominate early mitosis. While Figure 9A and 9B clearly suggest laterally interacting microtubules at the centromere, its not clear from the text if the authors are indeed referring to microtubule walls - this could be clarified as vast majority of prometaphase microtubule interactions are along walls and not via the ends of microtubules.

The manuscript text and figures are clearly presented. I include suggestions for minor text-edits that will improve the overall impact of the manuscript.

1. The authors indicate that "only microtubules localized close to the inner centromere would induce robust kinetochore phosphorylation, whereas microtubules that localize outside this region have little effect. " This model very nicely explains why localising AuroraB at the outerkinetochore disrupts monooriented prometaphase attachments but not bioriented end-on attachments - an unexplained observation (Shrestha 2017 Nat Com.) explained by their model could make their discussion stronger.

2. Page2: On Number of MTs are bound to human Kts. In the literature it seems like there is evidence for 20-40 microtubules binding to human KTs (Nixon 2015 eLife).

3. Page 8: "We conclude that Borealin microtubule-binding activity plays an important role in preventing and correcting improper kinetochore-microtubule attachments." I believe the evidence provided indicates Borealin's role in correcting improper attachments but not preventing improper attachments. This would not change the impact for their findings.

4. These sentences could be reworded:

"We treated cells depleted of Borealin and complemented with BorealinWT, Borealin MTBM or Borealin Δ 20 with 100nM paclitaxel and determined the duration of mitosis by live imaging (Fig.3A, B)."

Sup Figure legends: "Merge images include of LAPBorealin (green), ACA (Blue) and Hec1pS44 (grey)." "Merge images include of LAPBorealin(green), ACA (red) and Hec1 or Knl1 (grey)."

5. "recruitment of the phosphatase to "end-on attached" kinetochores31-34." 34 does not support this.

6. Scale bar description is occassionally missed out from figure legend.

Reviewer #2 (Remarks to the Author):

It is clear that the authors have painstakingly addressed all of my concerns. This paper represents an exciting study of CPC localization and dynamics and should be accepted for publication.

Reviewer #3 (Remarks to the Author):

The authors have addressed my comments fully.

Rebuttal to reviewer's final points:

1. The authors indicate that "only microtubules localized close to the inner centromere would induce robust kinetochore phosphorylation, whereas microtubules that localize outside this region have little effect. " This model very nicely explains why localising AuroraB at the outerkinetochore disrupts monooriented prometaphase attachments but not bioriented end-on attachments - an unexplained observation (Shrestha 2017 Nat Com.) explained by their model could make their discussion stronger.

We had to cut the discussion to get the manuscript under the word limit. However, the reviewer brings up a good point so we have added a new statement in the introduction that highlights this observation as a reason that it is important to understand how microtubule attachment is coordinated with Aurora kinase signaling.

2. Page2: On Number of MTs are bound to human Kts. In the literature it seems like there is evidence for 20-40 microtubules binding to human KTs (Nixon 2015 eLife).

The number changes on cell type and study between 17-40. We have used the number "around 20", which is a conservative estimate.

3. Page 8: "We conclude that Borealin microtubule-binding activity plays an important role in preventing and correcting improper kinetochore-microtubule attachments." I believe the evidence provided indicates Borealin's role in correcting improper attachments but not preventing improper attachments. This would not change the impact for their findings.
We have changed the line to "important role in preventing and/or correcting improper...." The reviewer is correct that we have only directly tested correction by washing cells out of monostral. However, many models in the field argue that the major role of the CPC is in prevention of incorrect attachments. If true, we would measure these in our assays in unperturbed cells. What is more important is that our model is applicable to both prevention and correction and therefore we would like to highlight this here.

4. These sentences could be reworded:

"We treated cells depleted of Borealin and complemented with BorealinWT, Borealin MTBM or Borealin Δ 20 with 100nM paclitaxel and determined the duration of mitosis by live imaging (Fig.3A, B)."

Sup Figure legends: "Merge images include of LAPBorealin (green), ACA (Blue) and Hec1pS44 (grey)." "Merge images include of LAPBorealin(green), ACA (red) and Hec1 or KnI1 (grey)."

Thank you we have made these changes.

5. "recruitment of the phosphatase to "end-on attached" kinetochores31-34." 34 does not support this.

Thank you we have deleted the reference.

6. Scale bar description is occasionally missed out from figure legend.

Thank you we have made sure that all of the figures have scale bars.